# Projected impacts of climate change on hydropower potential in China

Xingcai Liu[1], Qiuhong Tang[1], Nathalie Voisin[2], Huijuan Cui[3]

[1]Key Laboratory of Water Cycle and Related Land Surface Processes, Institute of Geographical Sciences and Natural Resources Research, Chinese Academy of Sciences, A11, Datun Road, Chaoyang District, Beijing, China.
[2]Pacific Northwest National Laboratory, 1100 N Dexter Ave, Seattle, WA, USA.
[3]Key Laboratory of Land Surface Pattern and Simulation, Institute of Geographical Sciences and Natural Resources Research, Chinese Academy of Sciences, 11A Datun Road, Chaoyang District, Beijing, China.

*Correspondence to*: Qiuhong Tang (tangqh@igsnrr.ac.cn)

**Abstract.** Hydropower is an important renewable energy source in China, but it is sensitive to climate change, because the changing climate may alter hydrological conditions (e.g., river flow and reservoir storage). Future changes and associated uncertainties in China's *gross* hydropower potential (GHP) and *developed* hydropower potential (DHP) are projected using simulations from eight global hydrological models (GHMs), including a large scale reservoir regulation model, forced by five general circulation models (GCMs) with climate data under two representative concentration pathways (RCP2.6 and RCP8.5). Results show that the estimation of the present GHP of China is comparable to other studies; overall, the annual GHP is projected to change by -1.7 to 2% in the near future (2020-2050) and increase by 3 to 6% in the late 21st century (2070-2099). The annual DHP is projected to change by -2.2 to -5.4% (0.7-1.7% of the total *installed* hydropower capacity [IHC]) and -1.3 to -4% (0.4-1.3% of total IHC) for 2020-2050 and 2070-2099, respectively. Regional variations emerge: GHP will increase in northern China, but decrease in southern China—mostly in South-Central China and Eastern China—where numerous reservoirs and large IHCs currently are located. The area with the highest GHP in Southwest China will have more GHP, while DHP will reduce in the regions with high IHC (e.g., Sichuan and Hubei) in the future. The largest decrease in DHP (in %) will occur in autumn or winter, when streamflow is relatively low and water use is competitive. Large ranges in hydropower estimates across GHMs and GCMs highlight the necessity of using multi-model assessments under climate change conditions. This study prompts the consideration of climate change in planning for hydropower development and operations in China, to be further combined with a socio-economic analysis for strategic expansion.

## 1 Introduction

Hydropower is a dominant renewable source of energy production and has received significant world-wide attention for further development (Ramachandra and Shruthi, 2007; Resch et al., 2008; Liu et al., 2011; Hamududu and Killingtveit, 2012; Stickler et al., 2013). China has a large gross potential, exceeded only by Russia (Zhou et al., 2015); hydropower provides about 17% of China's total electricity production (all technologies), and accounts for more than 80% of the nation's electricity energy

from renewable sources in 2012 (CNREC, 2013). A survey of hydropower resources showed that the gross hydropower potential (GHP, the total energy from all natural runoff at stream gradient over the entire domain) and the technically exploitable installed capacity (maximum possible hydropower generation) in China are 694 GW and 542 GW, respectively (Yan et al., 2006). Hydropower development in China has been greatly impelled by increasing environmental issues and energy demands (Huang and Yan, 2009; Liu et al., 2011; Lu, 2004). China's installed hydropower capacity (IHC) has grown by 11% per year during the past decade and reached 248 GW by 2012, which is about 46% of the technically exploitable potential of China (Liu, 2013; WEC, 2013). China will further foster its hydropower development in the near future (IEA, 2014) by targeting a total IHC of 350 GW in 2020, and most of it will be from the hydropower stations in Southwest China (GOSC, 2014).

Reduced hydropower generation has been reported to be associated with climate change (Qiu 2010; Bahadori et al., 2013), and significant progress has been made in assessing the impacts of climate change on hydropower elsewhere in the world. For example, it was reported that a future decrease in climate change-induced runoff would reduce energy generation and revenues of hydropower plants under current regulations in Columbia River and California hydropower systems in the USA (Hamlet et al., 2010; Vicuña et al., 2011). By assessing the hydropower system in the Peribonka River basin (Quebec, Canada), Minville et al. (2009) suggested that annual hydropower would decrease by 1.8% for the 2010-2039 period and then increase by 9.3% and 18.3% for the 2040-2069 and 2070-2099 periods, respectively. Considerable impact of climate change on hydropower was reported in the Swiss and Italian Alps regions, but the impacts varied for different locations, hydropower systems, and projections of climate change (Schaefli et al., 2007; Gaudard et al., 2014; Maran et al., 2014). Most studies suggested that new adaptive management may mitigate projected losses of hydropower in the Alps regions (Majone et al., 2016).

China is a large country (about 9.6 million km$^2$) and hydropower is and will be integrated in an electrical grid. Therefore a large scale assessment is required. Assessment of hydropower generation at large scales is more challenging because of the complex linkages of rivers and reservoirs. Lehner et al. (2005) estimated the developed hydropower potential (DHP) of existing hydropower stations, by assuming it to be a proportion of IHC, using the WaterGAP model and forcings from two general circulation models (GCMs) (HadCM3 and ECHAM4) under a moderate climate change scenario, and showed that the DHP would decrease by 7 to 12% over the entirety of Europe in the 2070s. Most recently, van Vliet et al. (2016) have used global macroscale integrated hydrologic modelling that includes a reservoir management model and projected reductions in the global annual hydropower capacities of 0.4 to 6.1% by the 2080s, based on the GCM-ensemble mean for representative concentration pathways (RCP2.6, 4.5, 6.0, and 8.5). Kao et al. (2015) found that federal hydropower generation will decrease by about 0.8 to 1.6% per year in the United States under a moderate carbon dioxide emissions scenario by regressing hydropower generation on streamflow. Bartos and Chester (2015) used similar regression analysis methods and found only small changes in hydropower generation capacity during the 2020-2060 period in 14 states in Western United States; however, they assumed potential adaptation of reservoir operations to future climate change conditions to keep constant head for hydropower

generation. This indicates that to some extent the impact of climate change on hydropower may be mitigated by changing the operational schemes of reservoirs (see also van Vliet et al. 2016). GHP addressed in those large-scale studies differs from DHP, because DHP looks at the expansion of the current hydropower fleet and how the hydropower potential could change. GHP affects the adaptation and mitigation planning for climate change. At large scales, changes in GHP were also paid great

attention (Zhou et al., 2015; Pokhrel et al., 2008; Lehner et al., 2005). For example, Lehner et al. (2005) showed that GHP would decline by 6% in Europe as a whole in the 2070s because of climate change. Zhou et al. (2015) estimated the global GHP by using runoff data derived from a global integrated assessment model (GCAM), and suggested that the total global GHP is approximately 128 PW ($10^{15}$ W) per year. Even though they did not estimate the changes under climate change, they concluded that current potential estimates of hydropower production are sensitive to regional variabilities such as climate,

population centres, i.e., future migration, and the economy, including the price of other electricity generation technologies. This finding highlights another source of uncertainty for planning and mitigation for hydropower under climate change conditions.

Runoff has experienced significantly decrease in the past decades and is likely to decrease more in many areas of China in the

future (Ma et al., 2010; Tang et al., 2013; Han et al., 2014; Sun et al., 2014; Schewe et al., 2014; Leng et al., 2015), which may significantly affect the water availability and the hydropower potential of rivers and at the current hydropower facilities. So far, most related studies have focused on the environmental and ecological impacts of the dams in China (Fan et al., 2015) or hydropower (potential) variations at country / continental level (Zhou et al., 2015; van Vliet et al., 2016), but the impacts of climate change on the hydropower of China are seldom reported as part of regional studies, i.e., including higher quality data

than what are available for global applications. It was partly due to the lack of continental hydrological simulations and necessary reservoir information at large scale. The Inter-Sectoral Impact Model Intercomparison Project (ISI-MIP) (Warszawski et al., 2014) provided multimodel hydrological projections over the world making it possible to investigate the implications of continental water resource changes under climate change. Therefore, it is of great interest to determine the impacts of future climate change on hydropower potential with respect to the underlying prosperous development of

hydropower in China.

This study aims to present a regional overview of China's hydropower potential under future climate change by using the projections from eight global hydrological models (GHMs) under two future climate scenarios provided by the ISI-MIP. Changes in GHP are estimated to quantify the impacts of climate change on the total hydropower capacity, and changes in

DHP (the developed hydropower potential of existing reservoirs) are estimated to examine the impacts on existing hydroelectric facilities through a hydropower scheme, which includes a reservoir operation module to regulate the simulated flow, similar to van Vliet et al. (2016). This study focus on the hydropower potential and the changes in reservoir hydropower capacity caused by the development of hydroelectric facilities are not considered because changes in reservoir operations are to be optimized across multiple objectives—water supply and flood control in particular—and are prone to coordination

between agencies and types of reservoir management (Tang et al., 2015). Nevertheless, this model-based analysis is expected to provide insight into future changes in current and additional potential hydropower generation of China, and to complement previous research studies at global scale (e.g. van Vliet et al., 2016): This study 1) assesses both gross and installed hydropower potential of China, and 2) provide an exhaustive uncertainty quantification with multimodel simulations and thus 3) supports regional development of China by focusing on regional variability. The presented modelling framework is compatible with integrated assessment models (IAMs) which can combine socio-economic analyses to further support the development of hydropower assets. This paper is organized as follows: Section 2 describes the method and data, Section 3 presents the results, Section 4 presents a discussion of the uncertainty associated with this study as well as the integration with socioeconomic analyses, and the last section presents the main conclusions.

## 2 Method and Data

### 2.1 Runoff and Discharge

Multimodel data are used to estimate GHP and DHP, and to address the uncertainty in the simulations. Daily runoff and monthly discharge in China are derived from eight GHMs: DBH (Tang et al., 2006, 2007b), H08 (Hanasaki et al., 2008), Mac-PDM.09 (Gosling and Arnell, 2011), MATSIRO (Takata et al., 2003), MPI-HM (Hagemann and Gates, 2003), PCR-GLOBWB (van Beek et al., 2011), VIC (Liang et al., 1994), and WBMplus (Wisser et al., 2010) provided by the ISI-MIP project (Warszawski et al., 2014). The model simulations are driven by the same forcing data downscaled from CMIP5 climate projections of the following GCMs: GFDL-ESM2M, HadGEM2-ES, IPSL-CM5A-LR, MIROC-ESM-CHEM, and NorESM1-M (Hempel et al., 2013). Hydrological simulations were all performed at a daily time step with a 0.5-degree latitude-longitude spatial resolution (~50 km at the Equator) over the 1971-2099 period. General descriptions of these GHMs and GCMs are listed in Tables S1 and S2. The use of different GHMs and GCMs allows for the evaluation of the uncertainties arising from the hydrologic and climate model structures in our estimates of GHP and DHP. Two RCPs, a low mitigation scenario (RCP2.6) and a very high baseline emission scenario (RCP8.5), are considered for representing the future climate and bounding the uncertainties in projections due to different RCPs. The model data have been used for assessment of the impact of climate change across several sectors (Davie et al., 2013; Elliott et al., 2014, Piontek et al., 2014, Schewe et al., 2014, Frieler et al., 2015). The overall GCM-GHMs framework are not validated in this study; the evaluations of the models are referred to the references listed in Tables S1 and S2.

### 2.2 Gross hydropower potential

GHP is defined as the total energy of natural runoff falling to the lowest level (e.g., sea level) of a specific region. GHP is estimated from discharge at each model grid cell: GHP = $Q \times h \times g$, where $Q$ is discharge estimated by the GHMs (m³/s); $h$ is the hydraulic head (m), i.e., elevation gradient in this case; and $g$ is gravitational acceleration (m/s²). A flow-routing scheme, following the river transport model in the Community Land Model (Oleson et al., 2010), is used to process the modelled runoff

into channel flow (runs in river) and cell-internal flow (runoff generated in the cell). For the discharge estimation in the GHP computation, flows are considered from both 1) cell-internal runoff that falls from the mean to the minimum elevation of the considered cell and 2) inflow that falls from the minimum elevation of the upstream cell to the minimum elevation of the considered cell. The separation of these two flows was proved to be more accurate for GHP estimation (Lehner et al., 2005).

## 2.3 Developed hydropower potential

DHP is defined as the maximum possible hydropower generation at the existing hydroelectric facilities, which refers to all reservoirs in this study. Therefore, DHP is not the actual hydropower generation of the current hydropower plants for the latter is affected by many socioeconomic factors such as energy demand, electricity price, various water uses, etc. DHP is estimated using a hydropower scheme based on reservoir information, such as location, storage capacity, dam height, and IHC, but also the transient flow passing through the turbines, including flow release constraints (environmental, spill) and transient variations in the hydraulic head, as explained. Generally, reservoir regulation could fairly consider both socioeconomic factors and climate change in assessing the hydropower of a single reservoir or small regions (e.g., Madani and Lund, 2010; Pérez-Díaz and Wilhelmi, 2010; Wu and Chen, 2011; Gaudard et al., 2013). However, though it is possible to develop a large-scale hydrological model linking energy and water based on a conventional reservoir regulation approach, calibration and validation of the modelling for individual reservoirs and then implementing a comprehensive assessment of hydropower at large scale remains challenging (Kao et al., 2015, van Vliet et al., 2016). Simplified, universal reservoir regulations were mostly adopted in large-scale hydrological modelling (Hanasaki et al., 2006; Döll et al., 2009; van Beek et al., 2011; Biemans et al., 2011) with limited regulation data and site-specific hydrological parameters.

In this study, socioeconomic factors (e.g., irrigation, energy price and demands) are generally not considered although other human water use is extensive in China (Tang et al., 2007a, 2008). The evaporation from water surface of reservoirs is neglected for this analysis because it represents a small fraction of the managed flow (Fekete et al. 2010; Liu et al., 2015). We use the generic reservoir regulation rules from Hanasaki et al. (2006) to derive the regulated flow at hydropower reservoirs. The reservoir operations scheme has been extensively used under different development efforts in large-scale studies (Biemans et al., 2011; Pokhrel et al., 2015; Döll et al., 2009) and at a more regional scale (Voisin et al., 2013; Hejazi et al., 2015). Regulation is set for flood control and then hydropower generation by targeting monthly releases for the wet and dry seasons in a year. That is, monthly release in a dry season is generally larger than monthly inflow of reservoirs; it thus gradually reduces reservoir storage for flood control in the coming wet season and provides more water for hydropower generation. Monthly release ($R_m$) is calculated as the case of no irrigation demands in Hanasaki et al. (2006):

$$R_m = \begin{cases} \left(\dfrac{c}{K_c}\right)^{\beta} k_y i_a + \left[1 - \left(\dfrac{c}{K_c}\right)^{\beta}\right] i_m \ , & 0 < c < K_c \\ k_y i_a \ , & c \geq K_c \end{cases} \tag{1}$$

where $K_c$ is the criterion of $c$, $i_m$ is monthly inflow (m³/s), $i_a$ is mean annual inflow (m³/s), $k_y = S_{beg}/\alpha C$, and $c = C/I_a$. $S_{beg}$ is the reservoir storage at the beginning of a year (m³), $C$ is the maximum storage capacity of reservoir (m³), $I_a$ is the mean total annual inflow (m³/yr), $\alpha$ is an empirical coefficient (0.85 in this study), which influences inter-annual variation in releases. The criterion of $c$ ($K_c$) is set as 0.5 and the exponent of ($c/K_c$), $\beta$, is set as 2 empirically, following Hanasaki et al. (2006). When the reservoir storage capacity is large compared to annual inflow ($c \geq K_c$), the monthly release is independent of monthly inflow and will be constant in a year if water is available. The reservoir is not allowed to release water when water storage is below 10% of the storage capacity, and the monthly release would be no less than 10% of mean monthly inflow for environmental flow. If water storage exceeds the storage capacity, excess water will be released.

The DHP of a reservoir is then estimated based on the monthly release (including spilling water): DHP = min($R_m \times h \times g$, IHC). Hydraulic head ($h$) is estimated by assuming it is linearly related to the reservoir storage, $h = S/A$, where $S$ is the mean reservoir storage at the beginning and the end of a calculation time step, and $A$ is the reservoir area. A is set as constant in this study, $A = C/H$, where $H$ is the dam height, which is also the maximum of $h$. The head $h$ is determined after the $R_m$ calculation. For the purpose of conservatively estimating the impact of climate change and representing a consistent estimate over multiple terrains, we use a cylinder shape for the reservoir to compute $h$ as a function of the storage. It differs from the tetrahedron shape used in Fekete et al. (2010) which represents reservoirs in complex terrain and will decrease the elasticity of the head with respect to changes in inflow.

447 reservoirs in China were selected from the Global Reservoir and Dam (GRanD) database (Lehner et al., 2011) if the key information of reservoirs are available. All the reservoirs were treated as hydropower plants and operations are determined with Equation 1. Run-of-the-river plants are not considered in this study for lack of hydropower station types in the current database. They presently do not represent a major capacity although it could change in the future for specific peak hour operations. Most reservoirs are located in East China (EC) and South-Central China (SCC), and a few of them are in North China. There are no IHC data associated with the GRanD reservoirs. The IHCs of provinces of China (about 130 GW in total) reported by CREEI (2004) are then used to determine the IHC of each reservoir as follows. Firstly, the provincial IHC is adjusted by the ratio of GRanD storage capacity to the reported storage capacity (National Bureau of Statistics of China, http://data.stats.gov.cn) for each province. Then the adjusted provincial IHC, which is about 114 GW in total, is proportionally arranged to powered reservoirs according to their storage capacity. Figure 1 shows the storage capacity determined from GRanD reservoir data, the reported storage capacity, and the reported IHC at 2004. Note that the IHC assignment is dependent on the relationship between storage capacity and IHC of all reservoirs, which may produce good estimates for some reservoirs (e.g. the Three Gorge Reservoir) but may also result in large errors if the IHC is not strongly correlated to storage capacity. The uncertainty resulted from the IHC assignment is discussed based on sensitive experiments (see section 4).

## 2.4 Experimental approach

GHP is determined from all GHM and GCM combinations over the historical period (1971-2000), and the ensemble median of gridded and regional annual mean GHPs are plotted in Figure 2. GHP distribution is dominated by the terrain and water availability showing high values in Southwest China and most South China, which is generally similar to previous studies at the global scale (Pokhrel et al., 2008; Zhou et al., 2015).

The analysis of GHP and DHP consists of two parts. In the first step, median estimates of the GHP and DHP over all GCM-GHM combinations are evaluated spatially and regionally over three periods: historical (1971-2000), near future (averages over the 2020-2050 period, referred to as 2035), and the end of the century (average of the 2070-2099 period and referred to as 2085, hereafter). A regional analysis is performed by evaluating changes between the two future periods and the benchmark historical period for the two RCPs. Emphasis is on regional change differences and uncertainties related to the two RCPs. The regions analysed are North China (NC), Northwest China (NWC), Southeast China (SEC), Southwest China (SWC), South-Central China (SCC) and Eastern China (EC), as shown in Figure 2. The interquartile range (IQR, the difference between 75% and 25% quantiles) of the multimodel ensembles is calculated to address the uncertainty of hydropower estimation between GCM-GHM combinations, and is shown in parenthesis following the median estimates.

A second analysis consists of looking at the temporal changes in the DHP and GHP over all of China and evaluating the overall trends in changes with respect to uncertainties related to inter-annual variability, and uncertainties related to individual GCM and GHM model structures. This analysis evaluates the changes in the DHP and GHP as time series. Estimates are computed as a 31-year moving average from 1971 to 2099 and labelled with the centre year. The labelled period is accordingly reduced to 1986-2084, and the labelled period of 2010-2084 is shown for a clear view.

To further support planning and mitigation for China hydropower, we define hydropower hotspots as regions with large hydropower potential or currently high IHCs. Two hotspot regions are isolated to illustrate the projected impacts of climate change on currently developed hydropower and hydropower that is in planning phases or under construction. One hotspot (*HS1*, see Figure 2) covers the areas with significant untapped hydropower potential in SWC, including most of the Jinsha River, Yalong River, Nu River, and Lancang River; many hydropower plants in this area are in the planning phases or under construction. The other hotspot region encompasses the Sichuan (including Chongqing) and Hubei provinces (*HS2*), and accounts for about 50% of total adjusted IHC in China in this study. We perform an additional analysis of expected changes over those two regions, which overlay parts and combinations of the main regions addressed in the first part of this paper.

## 3 Results

### 3.1 Gross hydropower potential

3.1.1 Validation

The ensemble median of the GHP of China is 644 GW (IQR: 200 GW), with a bias of -7.2% compared to the surveyed GHP

of 694 GW (Yan et al., 2006). Estimates of GHP show little differences in terms of percentiles between GCMs (medians range from 662 to 720 GW, see Table 1), but the IQRs across GHMs (more than 200 GW) are relatively large for all GCMs (see the inner plot (a) in Figure 2 and Table 1). This indicates that GHMs contribute a larger spread of GHP than GCMs over the historical period. Most GHP is located in southern China, especially in SWC (about 67% of total China GHP). The majority of hydropower resources are located in Yunnan, Sichuan, and part of Tibet in SWC, while a very small amount of resources

are located in large arid areas (e.g., desert) in NWC and NC. The EC and SCC regions are rich in water resources and presently have the largest portion of IHC (Figure 1), but most of them cannot produce as much hydropower as SWC because of their flatter terrains. The estimated regional GHP bias with respect to surveyed GHP is relatively large for different places (inner plot (b) in Figure 2), ranging from -38% (NWC) to +70% (NC), with the smallest (3%) being in EC. GHP is larger in summer (43%) and autumn (33%) than it is in spring (12%) and winter (7%) (Table 1).

3.1.2 Expected timeline of changes

Figure 3 shows the ensemble medians of China's seasonal and annual relative GHP changes under RCP2.6 and RCP8.5 for the 2010-2084 period. Relative changes are estimated by subtracting the historical GHP (i.e., GHP in Figure 2) from future periods then dividing by the historical GHP. As shown in Figure 3a, the annual GHP shows a small (<2%) decrease before

20 2020 and afterwards increases by nearly 3% under RCP2.6. Spring and summer GHPs generally show larger decreases (before 2035) and smaller increases than the annual GHP; the winter GHP shows little changes before 2060 and relative changes similar to the annual one thereafter. The autumn GHP shows a larger increase (about 8%) at the late 21st century than annual ones. The annual GHP under RCP8.5 would decrease by about 3% in 2020 and then increase after 2040. Annual, summer, and autumn GHPs would increase by 6% to 8% in the late 21st century, while winter and spring GHPs show small changes,

especially after the 2060. Annual GHP estimates often show a large spread across the GHMs and GCMs; e.g., the IQRs are as large as twice the median of the annual GHP in the late 21st century for both RCPs (see Tables S4 and S5).

3.1.3 Expected regional changes

Figure 4 shows the future relative annual GHP changes under RCP8.5. The total GHP of China is projected to decrease by

30 1.7% (IQR: 5.8%) in 2035 and increase by 6.3% (IQR: 13%) in 2085. The annual GHP shows smaller changes for both periods under RCP2.6: GHP is projected to increase by 2% (IQR: 5.7%) and 3% (IQR: 7.2%) in 2035 and 2085, respectively (see Figure S1, Tables S4 and S5). Estimations under RCP2.6 generally show spatial patterns similar to those under RCP 8.5 but

different magnitudes. For simplicity, we prefer to show RCP2.6 results in the Supplementary material, because the inter-regional analysis patterns are similar to RCP 8.5.

Under RCP8.5, annual GHP is projected to decrease by more than 10% in eastern SWC and NWC, and northern SCC, while in the Tibet region and most of NC, including the Tibet region, it is projected to increase by more than 5% in the near term (2035). The increase in annual GHP will dominate the relative changes in 2085, which shows a smaller decrease with respect to the historical period in SWC, SCC, and NWC, and larger increases in NC and western SWC. GHP changes might have significant implications in SWC, where more than half of China's hydropower structures are located and most hydropower stations are in planning phases or under construction (Wang et al., 2013). The projected annual GHP changes generally agree with the spatial patterns of discharge changes in China, i.e., they decrease in a large part of southern China and increase in most of NC (see Figure S2). In addition to annual changes in GHP, seasonal changes are computed in both 2035 and 2085 (see Figure S3 and Table S6). In the spring, the GHP will decrease on the southern edge of China, which is opposite to the trend of annual GHP at this location. Summer GHP changes resemble the annual ones, and relative changes in the autumn and winter GHPs seem to be larger than annual changes.

**3.2 Developed hydropower potential**

3.2.1 Projected timeline of changes

Figure 5 shows the estimation of DHP for existing reservoirs based on the present IHC of China (see Figure 1). The projected DHP shows a slower decrease in the late 21st century than in the near future. The annual DHP will decrease by 2.2% (RCP2.6) and 5.4% (RCP8.5) in 2035, and decrease by 1% (RCP2.6) and 3.6% (RCP8.5) in 2085. Under RCP2.6, the annual DHP will decrease by up to 3% by around 2025, and decrease by about 1 to 2% after 2050. Seasonal DHP shows changes similar to the annual one except the winter DHP decreases by about 4% in 2035. Under RCP8.5, the annual DHP will keep decreasing by more than 4% for most years after 2020. Seasonal DHP will keep decreasing in the future and will resemble the annual DHP, wherein summer shows a relatively smaller decrease and winter shows a larger decrease. However, the winter DHP is relatively small in a year. Therefore, the larger decrease in winter contributes little to annual DHP changes. Projected changes in the annual DHP also show large spreads across the GHMs and GCMs; e.g., IQRs/medians are greater than 2 for most years, especially for RCP2.6 (see Tables S7 and S8).

Note that the GHP would increase after 2040 for both RCPs, while the DHP would decrease more or less. This is mainly due to the fact that GHP mostly increases in NC and West China, where a small number of installed capacity plants are located; while GHP significantly decreases in SC and Central China, where a large number of installed capacity plants (expected large DHP) currently are located in this study (see Figures 1, 4 and S1).

3.2.2 Expected regional changes

Figure 6 shows the relative DHP changes of each reservoir in 2035 and 2085 under RCP8.5. More than 60% of the reservoirs show an expected decrease in DHP in 2035 (Figure 6a). Most DHP increases are less than 5% in China, and only several reservoirs show more than a 10% increase in NC. Most reservoirs in the southern EC and SCC show small change (less than 5%) in DHP, while importantly reservoirs that decrease by more than 10% are found in northern SCC and eastern SWC. Large decreases (more than 20%) in DHP mainly occur in northern SCC and EC, as well as in some reservoirs in SWC and NWC. The relative DHP changes in 2085 (Figure 6b) show a pattern similar to that in 2035, except that fewer reservoirs show large decreases in NWC and northern SCC, but more decreases are expected in the southern EC and SCC. Under RCP2.6, relative DHP changes generally resemble RCP8.5 patterns with larger increases in DHP in southern China and smaller decreases in DHP overall (see Figure S5). It seems that hydropower is more sensitive to discharge reduction for large reservoirs; e.g., DHP shows a relatively large decrease at most reservoirs with large storage capacities, but only small changes at many small reservoirs in SCC and SWC (Figure 6).

Figure 7 shows the regional relative changes in the aggregated monthly DHP in 2035 and 2085 under RCP8.5 (numbers are shown in Table 2). In 2035 (Figure 7a), the monthly DHP will decrease for most regions except for NC where it will increase slightly (less than 5%). The monthly DHP in NEC shows a very small increase from April to July, and decreases by about -0.2% (March) to -5.4% (October) in other months. The monthly DHP in EC shows slight changes from April to July and decreases considerably by 5% to 10% in other months. Changes in the monthly DHP in SCC resemble those in EC, and will decrease by up to 8.6%. The monthly DHP in both NWC and SWC will largely decrease during summer and winter, by up to 10% and 8%, respectively, but show small changes in other months. The monthly DHP of China will decrease more than 4.6% in all the months except May and June, which is similar to the DHP in SCC, and the annual DHP will decrease by about 5.4% (IQR: 7.4%), accounting for 1.7% of total IHC. The annual DHP shows small changes in NEC (-2.8%) and NC (1.4%), and shows the largest decrease in EC (-7.6%), followed by SCC (-5.4%) and NWC (-5.4%). As shown in the inner plot in Figure 7a, however, DHP changes in SCC (-1.1% of total IHC) contribute most to the changes in China, followed by EC (-0.4% of total IHC).

In 2085 (Figure 7b), the changes in monthly DHP are generally similar to those in 2035, except for a larger increase (4.8% to 10%) in NC, larger decreases in May (-6%) and June (-5.5%), and a small increase in other months (less than 3%) in NEC. The monthly DHP in EC will largely decrease (greater than 10%) in winter months, but there are fewer changes from May to July. The monthly DHP in SCC will slightly increase in June and decrease by 0.3% to 7.8% in other months. Both NWC and SWC show relatively small changes in spring and summer but large decrease in autumn and winter, except for May and June in NWC, which shows a large increase. Changes in the monthly DHP of China are very close to those in SCC; i.e., small changes in late spring and early summer and relatively large decreases (mostly greater than 5%) in other months. The annual DHP in China is expected to decrease by about 4% (IQR: 10.2%), accounting for about -1.3% of total IHC, which is also mostly contributed by DHP changes in SCC (-0.7% IHC) and EC (-0.4% IHC).

The monthly DHP changes in 2035 and 2085 under RCP2.6 (see Figure S6 and Table 2) are smaller than those under RCP8.5. Monthly DHP changes are mostly less than 5% and show more increases in NC, NEC, NWC, and SWC, especially in 2085; the annual DHP will decrease by 2.2% and 1.3%, accounting for 0.7% and 0.4% of total IHC, in 2035 and 2085, respectively.

**3.3 Impact on hotspot hydropower regions**

We now analyse the temporal trends over the two identified hydropower hotspots (HS1 and HS2) as defined in the experimental approach.

Figure 8a shows the relative changes in the monthly GHP and discharge of HS1 in 2035 and 2085. The annual GHP in HS1 is projected to increase by about 2% (IQR: 12%) in 2035, during which relatively large increases will occur in March, June, and from September to December, and considerable decreases will occur in April (-7.9%) and May (-5.5%). The mean flow will decrease for most months, which is not consistent with the GHP changes in some months. This suggests that the mean flow may decrease in an area with relatively low GHP (e.g., flat area). Figure 8b shows that the monthly GHP in 2085 will significantly increase by 0.4% (April) to 17% (September) and slightly decrease (-2%) in May, and the annual GHP will increase about 11% (IQR: 20.4%). Relative changes in discharge do not match the GHP changes exactly; e.g., the mean flow increases (13% to 23%) more than the GHP from May to August. GHP estimates in HS1 seem to have larger uncertainty in 2085 than in 2035. For RCP2.6 (see Figure S7a-b), the monthly GHP in HS1 will increase by 0.5% to 8% in 2035 and by 0.5 to 13% for most months in 2085. The annual GHP in HS1 will increase by about 6% (IQR: 8.9%) and 4.4% (IQR: 6.6%) in 2035 and 2085, respectively.

For HS2, the monthly DHP will significantly decrease by 3.3% (April) to 7.8% (August) for most months, but increase slightly in May (0.9%) and June (0.5%) in 2035 (Figure 8c). The monthly DHP in May and June will also increase by about 1% in 2085 (Figure 8d) and decrease by 1.2% (July) to 8.6% (November) in other months. The annual DHP will decrease by about 5.7% (IQR: 5.4%) and 5% (IQR: 10.7%), accounting for 1% and 0.9% of total IHC, in 2035 and 2085, respectively. Changes in the monthly mean inflow are smaller than or relatively close to those for DHP in 2085. This indicates that reservoir regulation may offset the impact of discharge changes on hydropower to some degree. Under RCP2.6, the DHP in HS2 shows similar but smaller changes than those under RCP8.5 (see Figure S7c-d). The monthly DHP will decrease by 0.5% to 5% in 2035 and by less than 2% for most months in 2085. The annual DHP will decrease by about 2.6% and 0.8%, accounting for 0.46% and 0.13% of total IHC, for 2035 and 2085, respectively.

For both hotspot regions, the uncertainties related to different GCM-GHM combinations are as large as the expected changes, but they are consistent in their monthly patterns and the direction of the annual changes for HS1 by the end of the century and HS2 for both the near term and the end of century.

## 4 Discussion

A selection of hydropower potentials (GHP, DHP) in this study was estimated using multimodel simulations of runoff and discharge under different climate change scenarios. An ensemble of hydropower potential is generated that represents uncertainties due to model structure (multi model) and emissions (RCPs). The combination of multiple potential estimates and the understanding of their regional diversity and associated uncertainties should provide support for integrated analysis from the generation perspective. A socio-economic analysis, discussed hereafter, would complement the decision support analysis from the water and energy demands perspective. In this section, we further discuss the uncertainties, how the results relate to the modelling framework and the socio-economic perspective.

Though the ensemble mean of projected GHP of China for the historical period is relatively close to the reported data, there is large discrepancy among GHMs. During the historical period, discrepancy in hydropower potential is much smaller among GCMs, because the GCM climate data is bias-corrected to a historical reference. It implies that validation or bias-correction may be helpful to reduce the uncertainty in the projections of GHMs. However, currently most GHMs except for a few such as WaterGAP are not calibrated against historical observations, and thus often show a large uncertainty in streamflow projections (Schewe et al., 2014). For annual estimates, it should be more effective and important to enhance the middle- and long-term hydrological prediction in order to fine tune the estimates of DHP and GHP. Therefore, validation and calibration of GHMs with hydrological observations (as the WaterGAP model did) are necessary in future studies which are effective in narrowing the differences among GHMs (Müller Schmied et al., 2014; Döll et al., 2016).

As stated, the uncertainty in the streamflow projections certainly propagates to the estimation of DHP. Though a universal reservoir regulation is applied to all modelled discharge, there is still a large spread across GCM-GHM combinations. The large uncertainty in DHP should be mainly due to the large discrepancy of GCM climate data since the reservoirs used in this study are mostly located in areas with low model agreements in future discharge projections (see Figure 1 in Schewe et al., 2014). This also partly explains why the total DHP (Figure 5) shows somewhat larger spread than the total GHP (Figure 3) of China.

The projections of hydropower potential in this study are generally consistent with previous studies (e.g. Zhou et al., 2016; van Vliet et al., 2016), but this analysis is extended for different potential development scenarios (gross and installed hydropower potential) and focuses on regional variability of China in the future. The uncertainty sources in the GHP and DHP estimates are further discussed as following, with respect to climate forcing, discrepancy among GHMs, reservoir regulation rules, and other sources like lack of consideration of socioeconomic factors in the hydropower estimation.

## 4.1 Uncertainties from models and climate forcings

The GHP estimates are representative of the effect of climate change on hydropower if all of the natural runoff could be captured. The effect of climate change on DHP is an intermediate estimate that takes into consideration the regulation of reservoirs as if they were operated for hydropower only. The combined direct and indirect impacts of climate change on hydropower can be very complicated, e.g., the more frequent extreme heat and drought may reduce power generation capacity in the future (Bartos and Chester, 2015) which may affect the electricity supply from the State Grid and therefore change the demand of hydropower generation. Nevertheless, the strong linkage between climate-streamflow-hydropower potential is one of the main ways that climate change affects hydropower. Climate change can directly modulate regional water availability, such as the increased temperature and depressed precipitation may give rise to draught events (Dai et al., 2004), while more intensive and spatially concentrated rainfall may result in more floods (Wasko et al., 2016). The consequent streamflow variations will then directly affect GHP and the reservoir storage which is associated with DHP. Besides the temperature and precipitation, several climate variables can alter river streamflow (e.g., Tang et al., 2013; Liu et al., 2014) and then affect hydropower potential. In this study, streamflow changes under changing climate conditions are projected by GHMs, and most of them include several climate variables, but two models use only temperature and precipitation as input (see Table S1). Therefore, this study presents the compound effects of the changes of multiple climate variables on hydropower potential. It is beyond the scope of this analysis to identify the contributions of all climate variables to the changes of hydropower potential. We limit the current analysis of meteorological forcing to a discussion on changes in temperature and precipitation, which are main drivers for streamflow projections.

As an example, the relative annual temperature and precipitation changes in 2035 and 2085 compared to the historical period are presented in Figures S8 and S9, respectively. Temperature is projected to increase by more than 1.5°C in most areas of China in the future under RCP2.6; it will increase by more than 1.5°C (2 °C) in southern (northern) China in 2035, and by more than 5°C in almost the whole of China in 2085 under RCP8.5. Annual temperature and precipitation increase a bit more in 2035 than in 2085 under RCP2.6, while temperature will largely increase in 2085 and precipitation will increase in most areas of China under RCP8.5. To a large extent, the spatial patterns of annual precipitation changes are in line with those of discharge changes in 2035, but they differ from the discharge changes in southern China in 2085, possibly due to significant warming.

The estimation of GHP is subject to large spread across the GCM-GHM ensemble, and most regions show poor agreement between models in the signs of the GHP changes (see Figure S4). The large uncertainty in climate projections (Knutti and Sedlacek, 2013) certainly will propagate to the hydropower estimates as GHMs are sensitive to climate forcing (Müller Schmied et al., 2014). The GCM uncertainty predominates in GHP estimates in this study. For both future periods, the spread owing to differences between GCMs dominates the total ensemble spread in most of southern China for RCP8.5 and RCP2.6

and in some of NEC for RCP2.6 (Figure S10). The ranges of GHP and DHP estimates across GHMs and GCMs are further summarized in Tables S4-S8, in terms of the 25th, 50th, and 75th percentiles of the ensemble of GCMs and GHMs, respectively.

The uncertainties in the DHP estimates, partially due to reservoir operations, are evaluated by looking at empirical parameters ($\alpha$, $\beta$, $K_c$) in Eq. (1) through several sensitivity tests (Table S3). The different values of these parameters represent different regulation efficiencies of reservoirs, and the experiments in Table S3 show the sensitivity of the DHP estimates to the regulation coefficients. Experiments with $\alpha$ = 0.65, 0.75, 0.85 (used in this study), 0.95, $\beta$ = 1, 2 (used in this study), 3, and $K_c$ = 0.4, 0.5 (used in this study), 0.6, are carried out, and one parameter value is changed for each experiment. These sensitivity tests show that estimates of DHP are considerably sensitive to the parameter $\beta$ (Figure S12), which could adjust the monthly release via the weights between annual inflow and monthly inflow. When $\beta$ = 3, the relative DHP changes would be smaller, and would have differences of about 2% from $\beta$ = 2. To some extent, this indicates that adjusting regulation rules may mitigate the impact of climate change on hydropower generation in the future. The DHP is also sensitive to the parameter $\alpha$ (Figure S11), which may affect the release coefficient ($K_y$); but it shows little sensitivity to the parameter $K_c$ (Figure S13). We also attempt to demonstrate the uncertainty in DHP estimates arising from the assignment of reservoir IHCs. Two experiments with IHC decreased (0.9*IHC) and increased (1.1*IHC) by 10% are further carried out (see Figure S14), and they show very little differences from the estimates of DHP using the present IHC. Nevertheless, more accurate IHC data are urgent needs for the future investigation of regional water resources-energy management under climate change.

In addition to the uncertainties in climate projections, GHMs, and regulation-related parameters, this study is also subject to considerable uncertainty in the estimates of DHP due to the simplification of reservoir regulation rules and limitations of the data. The different assumptions in the reservoir regulation are also a possible cause of the discrepancies compared to other studies (e.g., Fekete et al., 2010). The simplification of reservoir geometry for the computation of the hydraulic head may affect the DHP estimates in this study. More complex expression of the reservoir geometry may provide better approximation depending on the river geomography (Fekete et al., 2010) and then produce different DHP estimates. We further address here two main assumptions for estimating DHP: the effect of simple reservoir regulations and the lack of water withdrawals in the system for socioeconomic water demand.

### 4.2 Uncertainties associated with reservoir operations

Hydropower operations differ based on the reservoir storage and inflow characteristics, and if the plant is operated for general generation (daily load), or for capacity of generation during peak hour load. Furthermore, most reservoirs operated have multi-objectives to be combined with hydropower, implying even more complexities in hydropower operations. Large-scale water resources management modules have been efficient at evaluating the state of managed water resources over continents (Biemans et al., 2011). Although not operational models, the research models mimic reasonably well the impact of impoundment (regulation and withdrawals) on flow over large areas. Usually, two types of release rules are used. 1) The flood

control/hydropower rule mostly releases water uniformly over the year with inter-annual variability in the release and minimizes spilling. With a quasi-constant release target over one whole operational year, the storage buffers the seasonality in flow (for larger reservoirs), which affects DHP estimates. 2) An irrigation rule tops off the reservoir storage as much as possible before the start of the irrigation season, and then releases water with a monthly pattern following the monthly demand anomalies. In our regulation scheme, we only used the flood control/hydropower rule with no water or energy demand information. The estimates of DHPs in this paper are therefore an upper bound of hydropower generation and consistent with the concept of "potential" with respect to an operational context with more complex water management for competitive water uses, uncertainties in water demand.

Adaptation of reservoir operations to adapt to impact of climate change can be complex as they will need to be adjusted for both changes in water resources and increase competition between water uses (Vicuna et al., 2008, Vicuña et al., 2011; Finger et al., 2012; Jamali et al., 2013), and potentially to changes in energy demand and changes in energy infrastructure as well. In our analysis where the reservoir operations depend on the historical long-term mean annual inflow, an increase (decrease) in mean annual inflow into reservoirs as well as the change in seasonality affect the estimation of DHPs. As release targets are maintained, the change in storage head for large reservoir storage and reservoir spilling/drying are driving the estimates of DHP changes, No adaptation measure was applied to our reservoir operations as it would add another level of uncertainty to be further quantified and evaluated with respect to the competing water uses as addressed next.

### 4.3 Other sources of uncertainties

The projected changes in hydropower potential indicate the impacts of climate change, but they do not represent future prospects because socioeconomic and technical evolutions are not considered. Therefore, more uncertainty may arise: 1) Anthropogenic water use is expected to increase along with the increase in temperature and population (Kendy et al., 2007; Elliott et al., 2014; Leng and Tang, 2014), however, it is not considered. Socioeconomic water demand has been the focus of recent research in global integrated assessment and is taken into consideration in associated with the changes in population, industrial development, policy choices with respect to carbon emissions, economy between countries, and energy demands (Hejazi et al., 2014). 2) Changes in infrastructure should also be taken into consideration when estimating DHP. The South-North Water Diversion project, which was designed to divert about 44.8 billion $m^3$ water annually, would influence the hydropower generation potential in the Yangtze River (Zhang, 2009). 3) Hydropower generation is affected by its potential generation and by its integration into the electrical grid. The use of pumped-storage hydroelectricity (Huang and Yan, 2009), potential changes in seasonal energy demand and the electricity price of the power grid may affect the actual hydropower generation of a region or a reservoir. 4) Variation in climate change-induced energy demand may also affect actual hydropower generation; e.g., increasing temperature may lead to more energy demand in summer and less in winter (Pereira-Cardenal et al., 2014). These cumulative effects on hydropower merit further study and associated uncertainty quantification.

**4.4 Socio-economic implications from hydropower potential changes**

The development of hydropower assets is driven by the potential production of the plants/region as well as by its economic value, which in turns depends on the energy demand, distance between the demand and the generation, and prices of other electricity generation technologies (natural gas and coal for example). Other factors such as policy, technology development, electricity market, city expansion and industry location can affect the economic value of hydropower. This analysis supports the first step for developing hydropower assets (potential generation) and is driven by the natural resources conditions (climate, hydrology). It also provides an extensive uncertainty quantification. We discussed some sensitivity analyses and adaptation approaches. Beyond mitigation and adaptation in reservoir operations, a lower DHP would most likely need to be balanced by energy production from other sources, likely from costlier technologies, implying regional economic impact, which would need to be taken into consideration in the socio-economic analyses. The socio-economic analysis is beyond the scope of this paper. However, the regional assessment performed in this study provides the necessary information for integration into regional version of integrated assessment models (IAM) for national sustainable mitigation and adaptation policy making. It also provides a first assessment for regional developers to create case studies in specific sites to determine the feasibility from a natural resources and economical perspective.

**5 Conclusions**

An overview of projected future changes in the GHP and DHP of China was presented using hydrological simulations derived from multiple GHMs and GCMs. A reservoir regulation scheme was incorporated to estimate the DHP using current infrastructure. Historical GHP simulation was evaluated at the regional scale (overall bias -7.2%) and was generally close to the latest surveyed GHP in China. Most GHP is located in SWC, SCC, and NWC, where there are both rich water resources and large topography gradients. Projections of future changes in hydropower potential of China are generally consistent with previous studies (e.g. van Vliet et al., 2016). Two time slices, from 2020 to 2050 and from 2070 to 2099, were selected to further analyse the regional changes in China's GHP and DHP.

The GHP of China is projected to change by -1.7 to +2% in the near future (2020-2050), and increase by 3 to 6% by the late 21st century (2070-2099). Large regional variations emerge: a relatively large decrease will occur in SWC and SCC, especially in summer, and some increase will occur in most areas of NC.

The annual DHP in China will decrease by about 2.2 to 5.4% (0.7-1.7% of total IHC) and 1.3% to 4% (0.4-1.3% of total IHC) from 2020 to 2050 and from 2070 to 2099, respectively. These changes are mostly contributed by the large decrease in SCC and EC, where most reservoirs and large IHC are located currently. The DHP will decrease with some regional disparities as well. It will mainly decrease in southern China (e.g., EC, SCC, and part of SWC), and will increase considerably in NC and

the region of Tibet. China's DHP also shows a small decrease in late spring and early summer and a relatively large decrease in other months.

The impact of climate change on hydropower is particularly of concern in two identified hotspot regions that have rich hydropower potential. One hotspot located in SWC shows increases of nearly 2 to 6% and 4 to 11% in the annual GHP from 2020 to 2050 and from 2070 to 2099, respectively. This region has the most hydropower plants currently in planning phases or under construction, and will be the most important region for targeting hydropower development in China in the near future (GOSC, 2014). The result herein suggests the necessity of considering climate change for future hydropower development. In another hotspot region—the Sichuan and Hubei provinces—which holds nearly half of China's total IHC and is closer to urban centres, the DHP will decrease by 2.6 to 5.7% (0.46-0.97% of total IHC) and 0.8 to 5% (0.13-0.91% of total IHC) from 2020 to 2050 and 2070 to 2099, respectively. Though the DHP seasonality would be optimized (e.g. retain a high water level or increase release) to reduce the effects of monthly inflow decrease for certain years, it is mostly subject to the streamflow seasonality during a long-term period according to the parameterization of reservoir regulation. In this hotspot region, relatively small changes of monthly DHP will occur in late spring and early summer, while large decreases will occur in other months. If actual hydropower changes proportionally to the DHP under climate change, the reservoirs in this region might be unable to provide as much hydropower generation as present-day. The significant DHP decrease in dry season (e.g., in winter) will further increase challenges to managing the competitive water uses and regulation of reservoirs.

The projected effects of climate change on GHPs and DHPs of China are related but of opposite direction of change because most areas with high IHC show decrease in DHP while most areas with high GHP show increase in GHP. Even though GHPs are generally projected to increase by the second half of the 21st century, DHPs given the current infrastructure will not be able to mitigate the hydrological changes and thus will decrease without future update of regulation rules. Those trends tend to be consistent even under the range of uncertainty captured by an ensemble of global climate models, hydrological models, and two bounding climate change scenarios.

Hydropower is a relatively cheap and clean energy, which also facilitates the penetration of other renewable energy (solar, wind) into the grid. Both the numerous exploitable potential and the well-advanced technology of hydropower would greatly help China reduce its greenhouse gases emissions and environmental pollution while developing its economy (Kaygusuz, 2004; Chang et al., 2010; Hu and Cheng, 2013; Liu et al., 2013). It is necessary to involve climate risk assessment in hydropower development because the projected decrease in flow under climate change conditions will lead to reduced potential hydropower generation with high regional disparities. China has shown strong motivation to develop large hydroelectric facilities in the future. This motivation is not only driven by hydropower energy but also by flood control or irrigation demands. This research provides a preliminary regional assessment of climate change impacts on hydropower potential, which could guide the

development of hydropower technology, e.g., pumped-storage hydroelectricity, inter-basin transfer, joint reservoir operations, in order to mitigate the impact of climate change on renewable electricity generation in China.

*Author contributions*. Q. Tang and X. Liu designed the research, N. Voisin processed the reservoir data, X. Liu conducted analyses, Q. Tang, X. Liu, N. Voisin, and H. Cui wrote the manuscript.

*Acknowledgements*. We acknowledge the modelling groups (listed in Table S1 of this paper) and the ISI-MIP coordination team for providing the model data. This research is supported by the National Natural Science Foundation of China (41425002 and 41201201) and the National Youth Top-notch Talent Support Program in China.

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

**Tables**

Table 1. Estimates of annual and seasonal GHP (GW) for regions and China over 1971-2000.

Table 2. Percentiles of annual DHP changes (%) for regions in China over 2020-2050 (2035) and 2070-2099 (2085) across the ensemble of GCM-GHM combinations.

Table 1. Estimates of annual and seasonal *GHP* (GW) for regions and China over 1971-2000.

| Percentile / GHM | 50th | 25th | 75th | IQR |
|---|---|---|---|---|
| GFDL-ESM2M | 720 | 510 | 743 | 234 |
| HadGEM2-ES | 676 | 482 | 710 | 228 |
| IPSL-CM5A-LR | 662 | 490 | 690 | 200 |
| MIROC-ESM-CHEM | 715 | 506 | 734 | 228 |
| NorESM1-M | 704 | 487 | 755 | 268 |
| | | | | |
| **Region** | | | | |
| NC | 25 | 11 | 32 | 21 |
| NEC | 11 | 9 | 14 | 5 |
| EC | 29 | 26 | 31 | 5 |
| SCC | 91 | 79 | 100 | 21 |
| SWC | 440 | 356 | 488 | 132 |
| NWC | 62 | 40 | 74 | 34 |
| | | | | |
| **Season** | | | | |
| MAM | 314 | 280 | 385 | 105 |
| JJA | 1116 | 941 | 1218 | 278 |
| SON | e | 662 | 914 | 252 |
| DJF | 189 | 125 | 270 | 146 |

Table 2. Percentiles of annual *DHP* changes (%) for regions in China over 2020-2050 (2035) and 2070-2099 (2085) across the ensemble of GCM-GHM combinations.

| Percentile Region | RCP2.6 | | | RCP8.5 | | |
|---|---|---|---|---|---|---|
| | 50th | 25th | 75th | 50th | 25th | 75th |
| **2035** | | | | | | |
| North | 7.03 | 2.65 | 11.56 | 1.42 | -3.73 | 12.48 |
| Northeast | 2.15 | -2.83 | 9.19 | -2.83 | -11.02 | 3.08 |
| East | 1.89 | -11.18 | 3.10 | -7.57 | -11.69 | -0.83 |
| South Central | -3.26 | -5.51 | -0.42 | -5.39 | -9.11 | -2.78 |
| Northwest | -2.78 | -4.98 | 1.47 | -5.42 | -7.00 | -0.92 |
| Southwest | -3.41 | -9.08 | 1.75 | -4.56 | -13.23 | 0.25 |
| Hotspot 2 | -2.55 | -4.92 | -0.43 | -5.72 | -8.41 | -3.03 |
| China | -2.22 | -4.91 | 0.33 | -5.44 | -8.95 | -1.46 |
| | | | | | | |
| **2085** | | | | | | |
| North | 3.52 | -1.84 | 12.56 | 7.18 | -0.20 | 15.44 |
| Northeast | 0.40 | -6.12 | 9.37 | -1.84 | -13.02 | 15.03 |
| East | -2.35 | -7.29 | 3.23 | -7.07 | -11.81 | 2.60 |
| South Central | -1.07 | -4.49 | 2.70 | -3.77 | -9.75 | 0.59 |
| Northwest | 0.14 | -3.65 | 3.95 | -3.41 | -8.57 | 1.86 |
| Southwest | 1.64 | -4.65 | 4.52 | -0.48 | -14.96 | 9.77 |
| Hotspot 2 | -0.82 | -3.61 | 2.90 | -5.01 | -9.60 | 1.13 |
| China | -1.25 | -4.93 | 2.57 | -3.85 | -10.47 | 0.24 |

**Figures**

Figure 1. Reservoir storage capacity and installed hydropower capacity (IHC) at the provincial level in mainland China.

Figure 2. Medians of the GHPs of China across the ensemble of all GCM-GHM combinations over the historical period (1971-2000). Red rectangular denotes a hotspot region (HS1). NC: North China, NEC: Northeast China, EC: East China, SCC: South Central China, NWC: Northwest China, SWC: Southwest China. Inner plot (a) shows the boxplot of GHP of China across GHMs for each GCM, where the red line is the reported GHP, i.e. 694GW, and G, H, I, M, and N denote GFDL-ESM2M, HadGEM2-ES, IPSL-CM5A-LR, MIROC-ESM-CHEM, NorESM1-M, respectively. The inner plot (b) shows the boxplot of regional GHPs across all GCM-GHM combinations on a log scale, where red dots denote reported GHPs (Li and Shi, 2006).

Figure 3. Medians of relative changes in the GHP of China over the 2010-2084 period under RCP2.6 (a) and RCP8.5 (b). Solid lines show the ensemble medians of GHP; grey areas denote the IQRs of annual GHP. MAM: May, April, May; JJA: June, July, August; SON: September, October, November; DJF: December, January, February. All the annual and seasonal time series of GHP are estimated at a 31-year moving average over 1971-2099 and labeled with the center year.

Figure 4. Medians of relative changes in the annual mean GHPs for 2020-2050 (a) and 2070-2099 (b) compared to the historical period (1971-2000) across the ensemble of GCM-GHM combinations under RCP8.5.

Figure 5. Medians of relative changes in the DHPs of present reservoirs in China over the 2010-2084 period under RCP2.6 (a) and RCP8.5 (b). Solid lines show the ensemble medians of DHP; grey areas denote the IQRs of the annual DHPs. All of the annual and seasonal time series of the DHPs are estimated at a 31-year moving average over 1971-2099 and are labeled with the center year.

Figure 6. Medians of relative DHP changes for present reservoirs in China across the ensemble of GCM-GHM combinations for 2020-2050 (a) and for 2070-2099 (b) under RCP8.5. Black lines depict the hotspot region HS2; i.e., Sichuan (including Chongqing) and Hubei province. Circle size is determined according to the logarithm of reservoir storage capacity.

Figure 7. Relative monthly DHP changes in regions in China for 2020-2050 (a) and 2070-2099 (b) under RCP8.5. Lines show the ensemble medians across all GCM-GHM combinations; grey areas show the IQR of relative DHP changes in China; the inner plots show annual DHP changes in regions in terms of the percentage of IHC.

Figure 8. Relative changes in the monthly GHPs (DHPs) and discharges (reservoir inflow) in the hotspot regions for 2020-2050 (2035) and 2070-2099 (2085) under RCP8.5. HS1: the hotspot region in Southwest China (see Figure 2); HS2: Sichuan and Hubei provinces (see Figure 6). Grey areas denote the IQRs across the ensemble of GCM-GHM combinations.

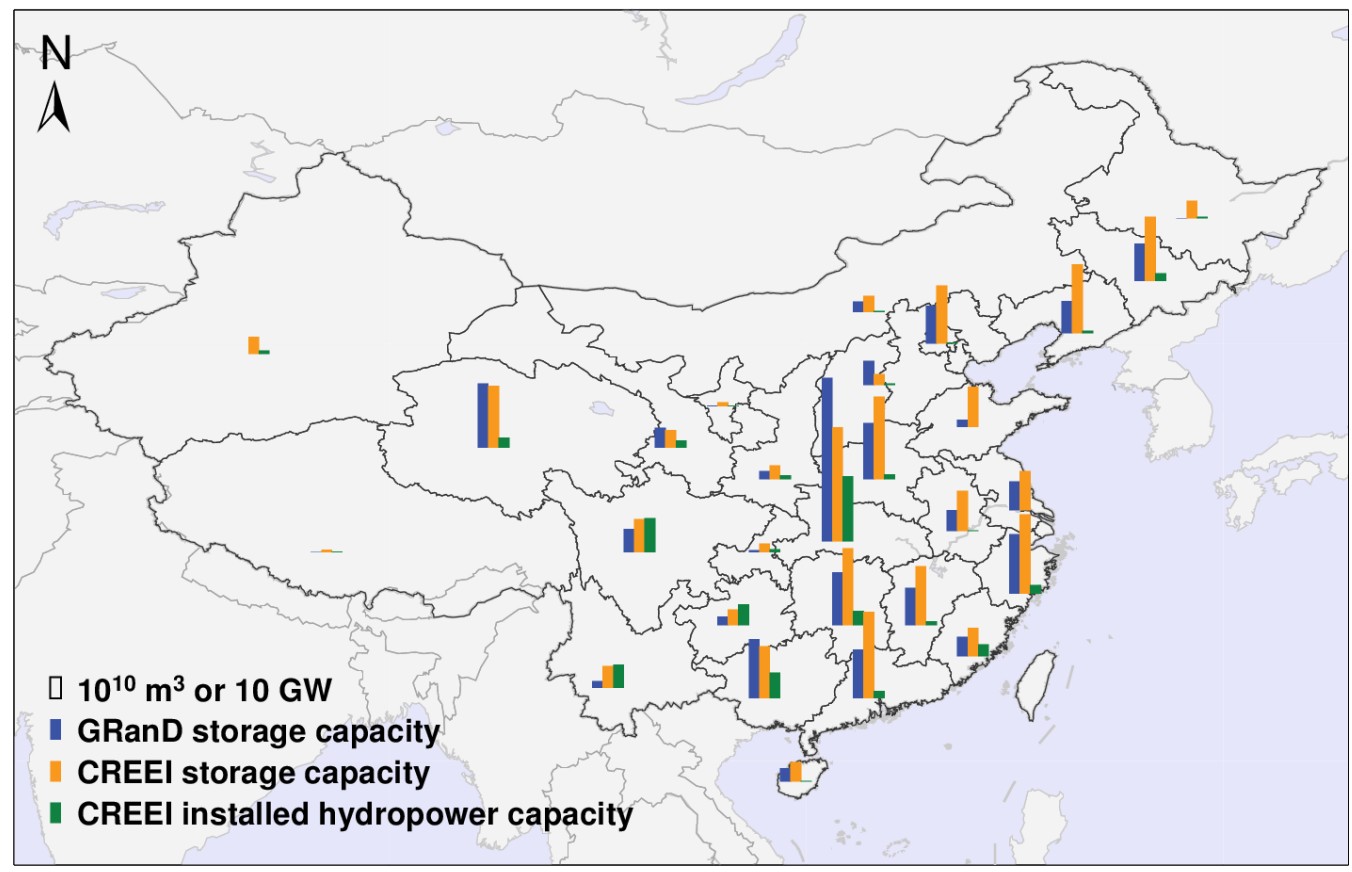

Figure 1. Reservoir storage capacity and installed hydropower capacity (IHC) at the provincial level in mainland China. GRanD refers to the Global Reservoir and Dam database, and CREEI refers to the reference CREEI (2004).

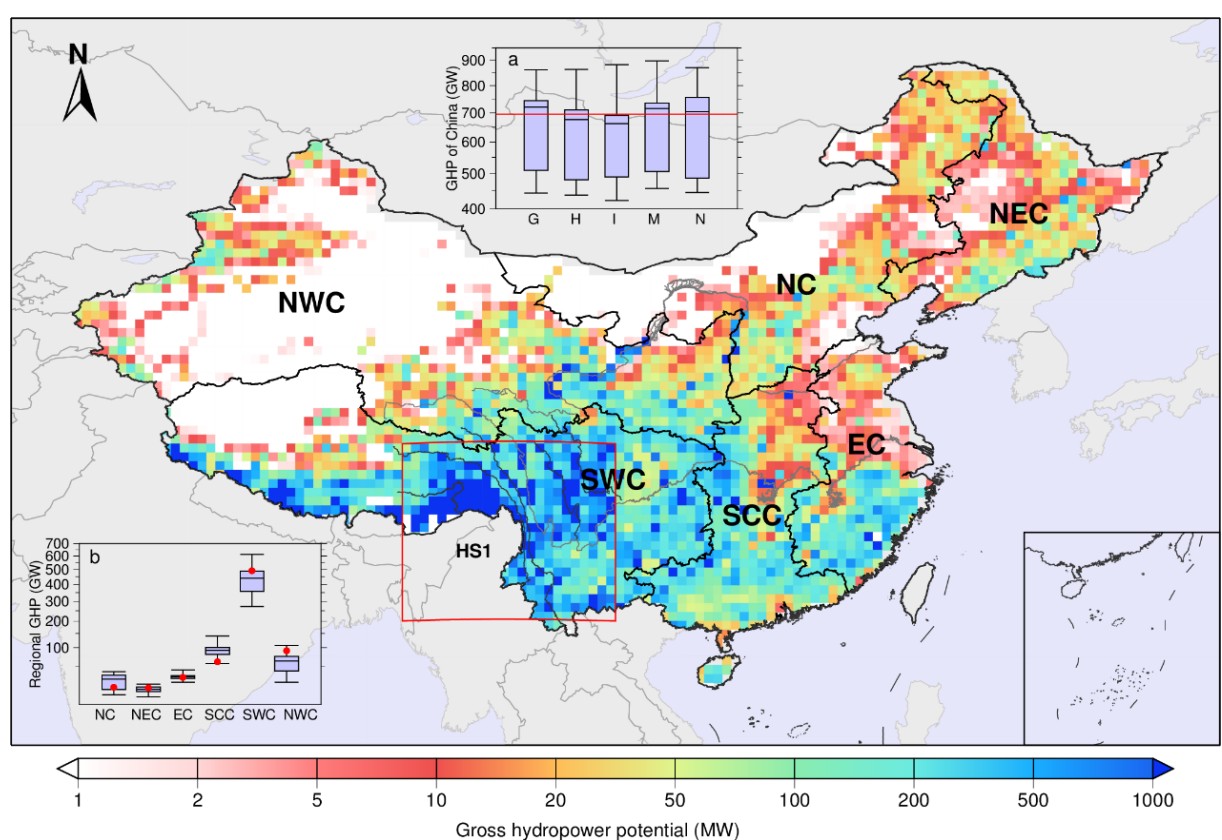

Figure 2. Medians of the GHPs of China across the ensemble of all GCM-GHM combinations over the historical period (1971-2000). Red rectangular denotes a hotspot region (HS1). NC: North China, NEC: Northeast China, EC: East China, SCC: South Central China, NWC: Northwest China, SWC: Southwest China. Inner plot (a) shows the boxplot of GHP of China across GHMs for each GCM, where the red line is the reported GHP, i.e. 694GW, and G, H, I, M, and N denote GFDL-ESM2M, HadGEM2-ES, IPSL-CM5A-LR, MIROC-ESM-CHEM, NorESM1-M, respectively. The inner plot (b) shows the boxplot of regional GHPs across all GCM-GHM combinations on a log scale, where red dots denote reported GHPs (Li and Shi, 2006).

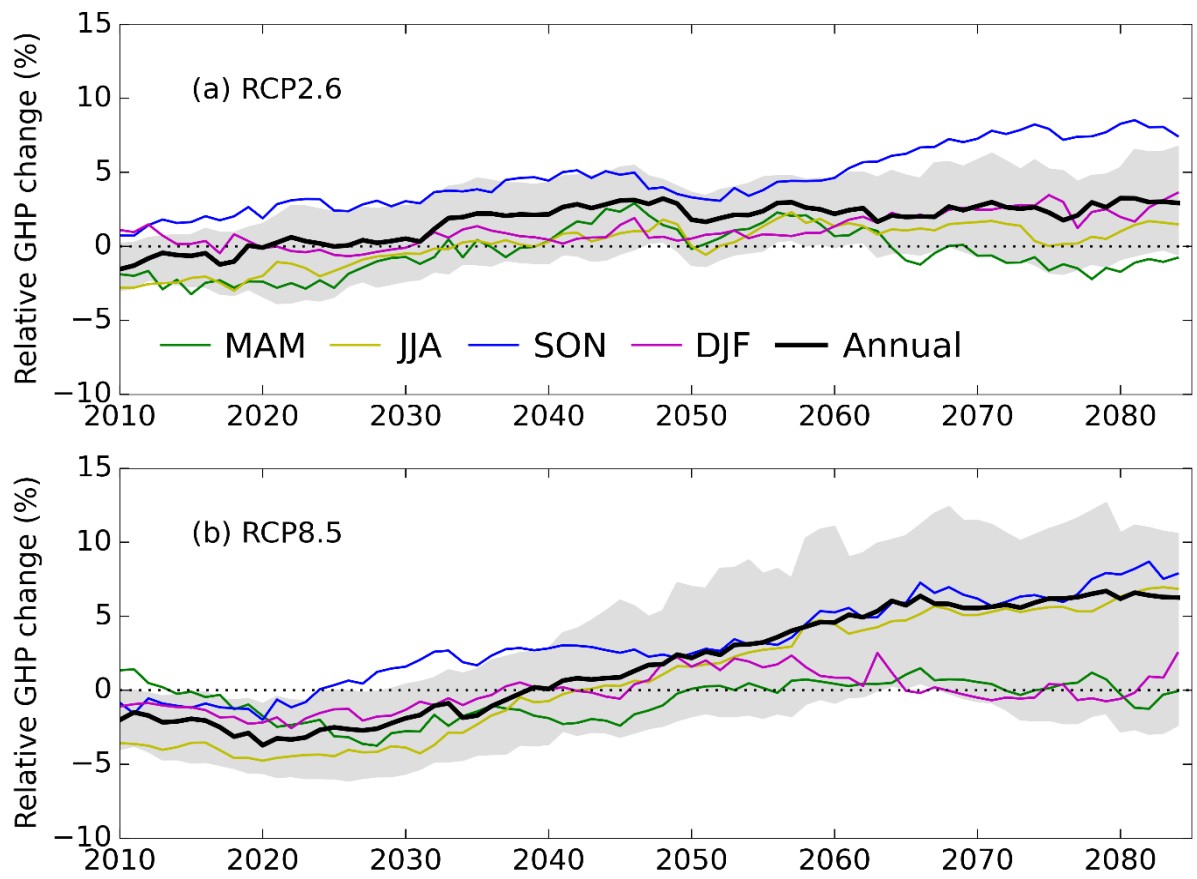

Figure 3. Medians of relative changes in the GHP of China over the 2010-2084 period under RCP2.6 (a) and RCP8.5 (b). Solid lines show the ensemble medians of GHP; grey areas denote the IQRs of annual GHP. MAM: May, April, May; JJA: June, July, August; SON: September, October, November; DJF: December, January, February. All the annual and seasonal time series of GHP are estimated at a 31-year moving average over 1971-2099 and labeled with the center year.

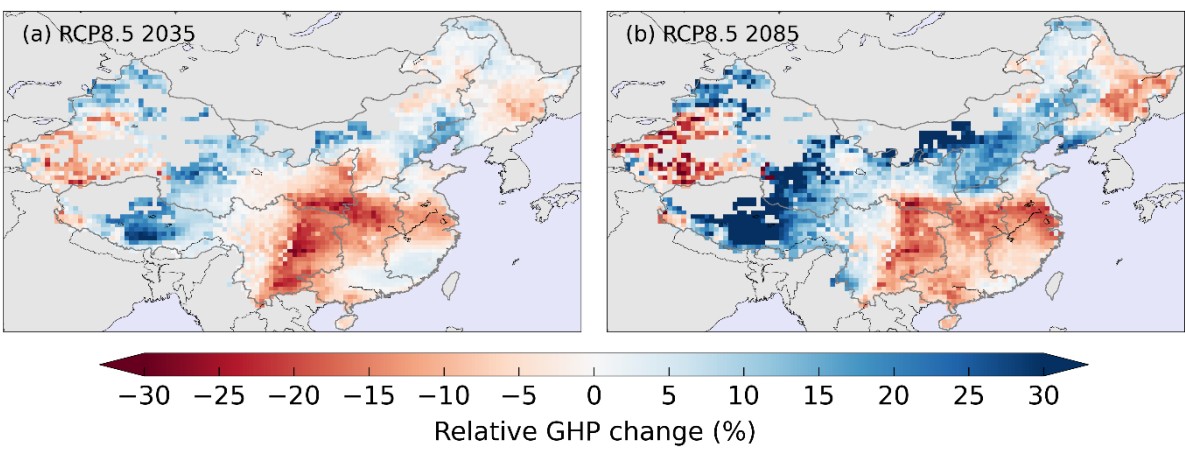

Figure 4. Medians of relative changes in the annual mean GHPs for 2020-2050 (a) and 2070-2099 (b) compared to the historical period (1971-2000) across the ensemble of GCM-GHM combinations under RCP8.5.

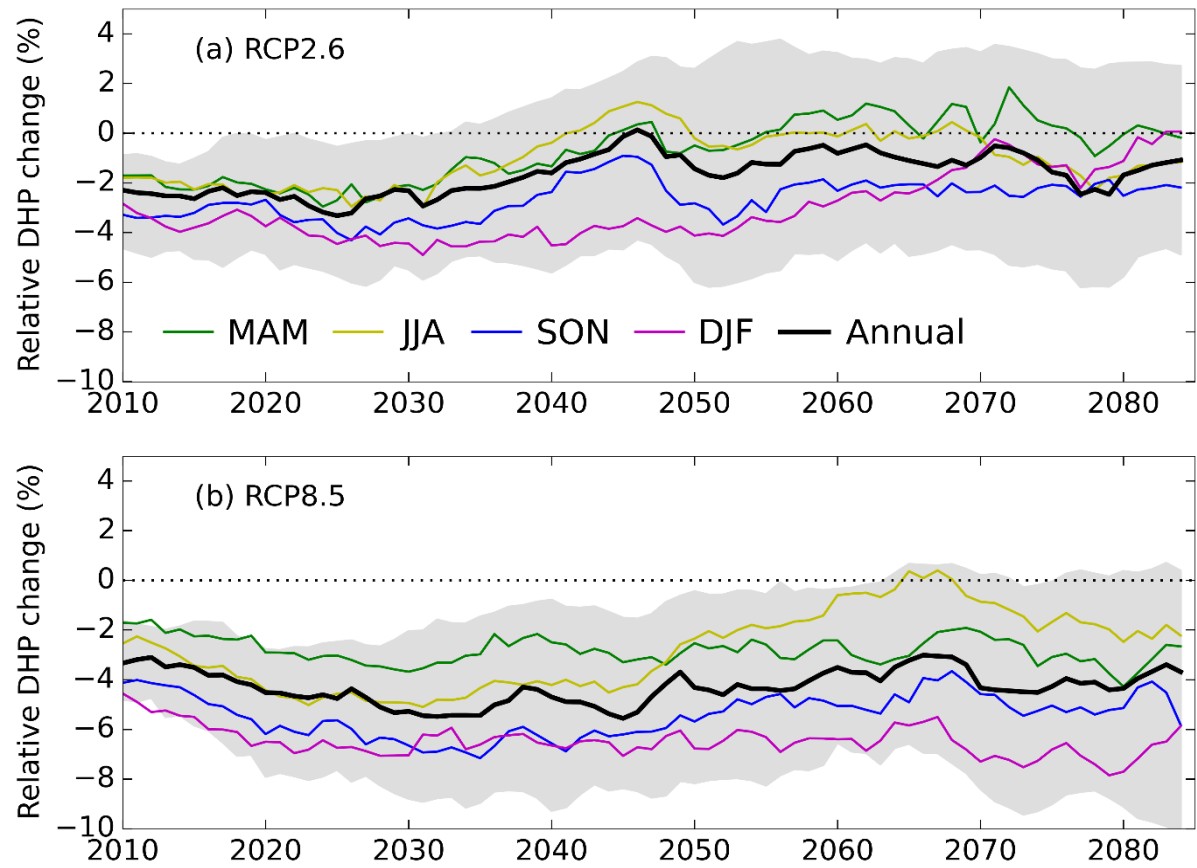

Figure 5. Medians of relative changes in the DHPs of present reservoirs in China over the 2010-2084 period under RCP2.6 (a) and RCP8.5 (b). Solid lines show the ensemble medians of DHP; grey areas denote the IQRs of the annual DHPs. All of the annual and seasonal time series of the DHPs are estimated at a 31-year moving average over 1971-2099 and are labeled with the center year.

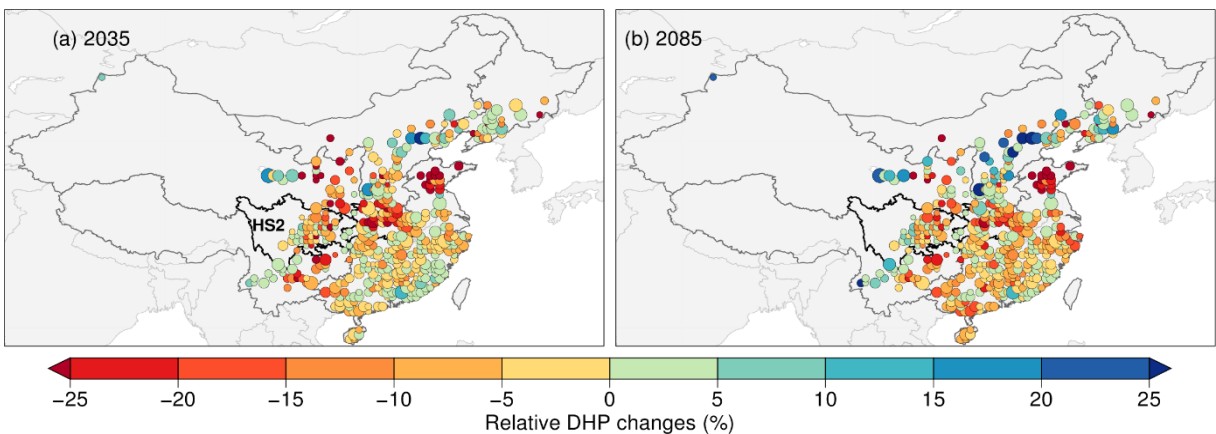

Figure 6. Medians of relative DHP changes for present reservoirs in China across the ensemble of GCM-GHM combinations for 2020-2050 (a) and for 2070-2099 (b) under RCP8.5. Black lines depict the hotspot region HS2; i.e., Sichuan (including Chongqing) and Hubei province. Circle size is determined according to the logarithm of reservoir storage capacity.

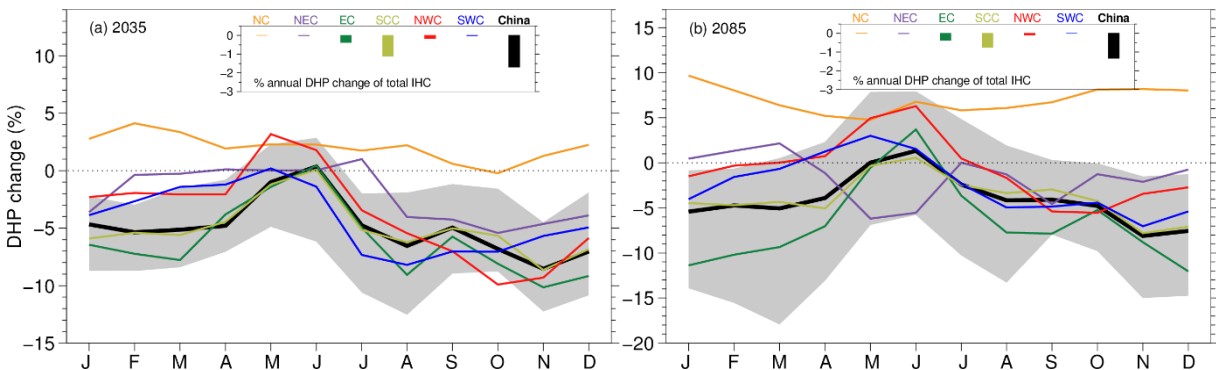

Figure 7. Relative monthly DHP changes in regions in China for 2020-2050 (a) and 2070-2099 (b) under RCP8.5. Lines show the ensemble medians across all GCM-GHM combinations; grey areas show the IQR of relative DHP changes in China; the inner plots show annual DHP changes in regions in terms of the percentage of IHC.

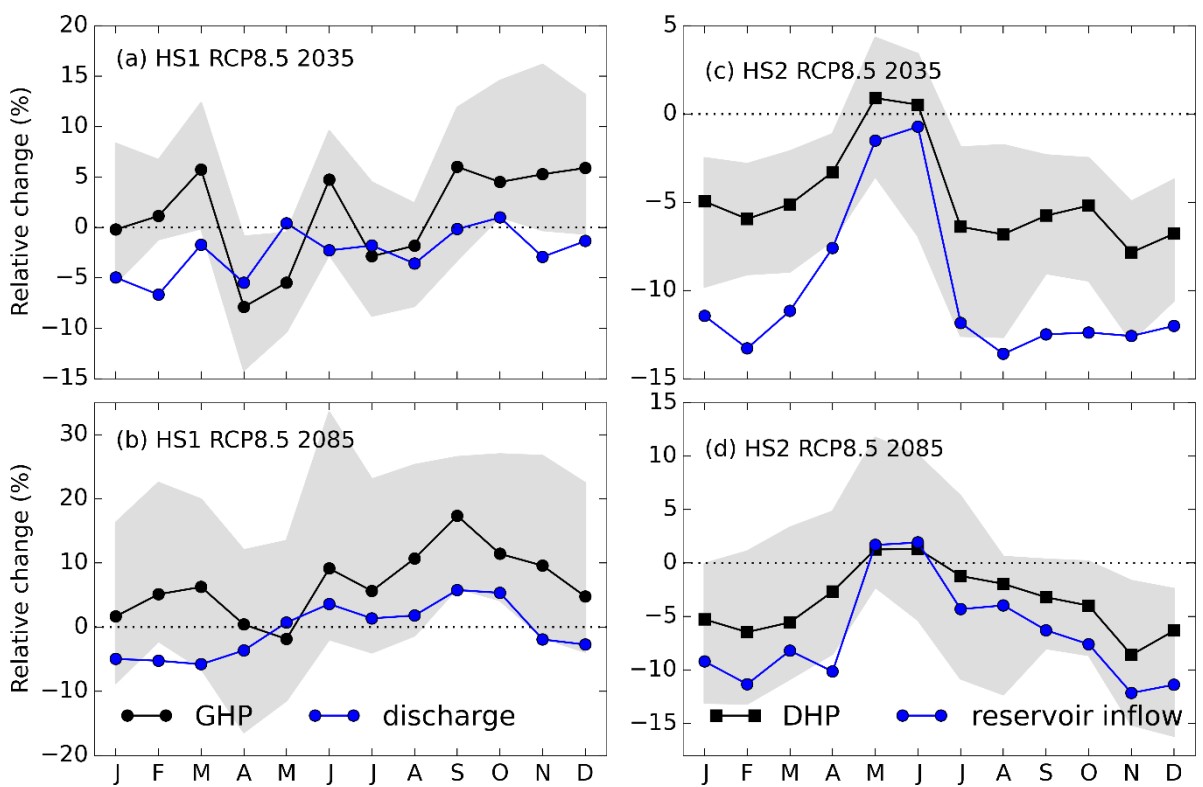

Figure 8. Relative changes in the monthly GHPs (DHPs) and discharges (reservoir inflow) in the hotspot regions for 2020-2050 (2035) and 2070-2099 (2085) under RCP8.5. HS1: the hotspot region in Southwest China (see Figure 2); HS2: Sichuan and Hubei provinces (see Figure 6). Grey areas denote the IQRs across the ensemble of GCM-GHM combinations.

