# Peer review of "Projected impacts of climate change on hydropower potential in China"

_Hydrology and Earth System Sciences, 2016_

## Referee Comment (RC1) · Anonymous Referee #1 · 8 Apr 2016

Review summary:

This manuscript uses multiple global hydrological models driven by multiple climate model data for two representative concentration pathways (RCPs) to estimate China's hydropower generation potential and the projected future changes based on the river flow estimated by these hydrological models. The study finds that the estimated present-day gross hydropower potential of China is comparable to previous estimates, and suggests that the hydropower potential will decrease in the short-term but will increase by the late 21st century. The study also suggests that these changes vary significantly across different regions. The results presented are of high interest to the scientific community and beyond as the global society today is increasingly concerned about the use of carbon-intensive energy sources to meet the rising energy demands and hydropower could potentially play an important role in future energy mix toward

reducing emissions and mitigating climate change, particularly in the rising economies such as China. Therefore, there is no doubt that the paper addresses an important topic but I feel that the study could be driven more by a central scientific finding with important socio-economic implications, rather than just presenting the changes in hydropower potential across different regions.

Specific comments:

(1) I suggest the authors to revise the introduction. The first paragraph doesn't read very well. Also, it is important to highlight the objectives of the study and the key questions addressed at the end of introduction.

(2) While the gross generation potential provides useful information on the potential future changes, it is not an indicator of actual power generation potential. So, it will be important to consider whether the available flows can be utilized to the fullest as well as various locational and technological constraints. The study doesn't provide any information on this aspect.

(3) Moreover, the analysis low flows would provide further insights on how the run-off-the-river hydropower generation capacity would be affected in the future. The annual mean and seasonal changes do not necessarily reflect such effects unless all runoff will be captured in reservoirs.

(4) In page 4, line 2 it is noted the reservoir module is similar to the one in van Vliet et al. (2016). What are the differences in the findings? It may be worthwhile highlighting the differences.

(5) Page 5, Line 25: Do all models use the same reservoir operation module?

(6) Page 6, Line 24: Change "great" to "high".

(7) Page 6, Line 12: Why and how were these 447 reservoirs selected?

(8) Page 7, Line 19: Expand this section or delete this line.
(9) Section 3: I see that a lot of information is provided as supplementary material. For completeness, I suggest the authors to bring some of these tables to the manuscript itself.

(10) Page 7, Line 27: change "is" to "are"

(11) Page 12, Line 11: Change "great" to "large"

(12) What is the rationale behind the use of different alpha, beta, and K values? This needs to be discussed in relation to the implications on results.

(13) Evaporation from water retention behind large dams could increase largely under warmer future climate which can reduce runoff. Is this considered in the present study?

---

## Referee Comment (RC2) · Anonymous Referee #2 · 12 Apr 2016

General comments

The authors projected Gross Hydropower Potential (GHP) and Developed Hydropower Potential (DHP) of China using the global runoff dataset developed by the ISI-MIP project. The dataset includes global gridded runoff field simulated by 8 global hydrological models for 5 climate models and 2 emission scenarios. They analyzed the spatial and temporal distribution of changes in GHP and DHP in China. Although hydropower is a fundamental source of energy, analyses utilizing macro scale hydrological model have been seldom reported. This report has potential to advance this research field.

As is commonly seen in macro scale hydrological simulations, this study is based on several strong assumptions. I have not been fully convinced by the validity of some of these assumptions. This is partly due to the assumptions themselves, but largely due

to lack of discussion. Details are noted below.

First, overall discussion on the background mechanism for the results is lacking. The results the authors obtained are well presented, but why and how they were obtained is little described. The Discussion Section should be largely expanded to include the mechanisms. Second, the term (and the model) of DHP should be revisited. What does "developed potential" mean? Which is more close to hydropower generation or technical hydropower potential? If DHP is different from any important indicators in the real world, how should we interpret the results? Without clarification of DHP, it is not clear what was calculated and what for.Third, the quality in runoff field of ISI-MIP should be well discussed. Since the global hydrological models participated in ISI-MIP have not been calibrated except the WaterGAP model, it must be carefully discussed that how the biases in runoff propagate to the results. Fourth, as far as I understood, the authors assigned the national total Installed Hydropower Capacity (IHC) into 447 major reservoirs. Since this might significantly overestimate IHC at individual reservoirs, the validity of this treatment should be validated and discussed. It might be a good idea to start with comparing reported installed hydropower capacity at individual reservoirs with the authors' estimation.

Specific comments

Page 6 Line 7," DHP = min (Rm x h x g, IHC)": I found that this equation primarily expresses hydropower generation. Why was this termed "Developed Hydropower Potential", not hydropower generation? If DHP is not hydropower generation, then what is this correspond to in the reality?

Page 6 Line 8 "h=S/A": Fekete et al. (2010) expressed reservoirs as tetrahedrons in their model. What are the advantage and disadvantage of the authors' expression (cylinder)?

Page 6 line 15 "no IHC data associated with the GRanD reservoirs" World Register of Dam by International Commission of Large Dams (http://www.icold-cigb.org/) includes

Electric Capacity of individual dams.

Page 6 line 18 "Then the adjusted provincial IHC..." As far as I understand, this study deals with storage and discharge for 447 reservoirs in China, while IHC for all the nation. This discrepancy could make Rm x h x g substantially smaller than IHC, hence it may have influenced the results. This point should be clarified here.

Page 11 Line 22. I got a general impression that the Discussion Section is superficial. Since the Results Section only introduces the numbers that authors obtained, actually I expected detailed discussion on the background mechanisms of model behaviors and interpretation of the results, but these are seldom provided in the current form of the manuscript. The contents of this section should be substantially added.

Page 12 Lne 10 "most regions show poor agreement between models": In terms of what? Magnitude or signs? What are the results of the WaterGAP model or the only model with calibration?

Page 13 Line 19 "Thus, reservoir regulation could be changed in the future to adapt to climate change": Too superficial and abstract. How should it be changed based on the findings of this study?

Page 15 Line 5 "Relatively small changes also will occur in late spring and early summer, while large decreases will occur in other months". Why did these happen in your simulations? Basic mechanisms should be mentioned here. For instance, DHP is a function of monthly discharge (Rm) and water level (h). Which is dominant factor to produce the seasonal variation?

Page 15 Line 10 "DHPs given the current infrastructure will not be able to mitigate the hydrological changes and thus will decrease": Why and how did the authors conclude this? Would this conclusion be different if the authors modified the reservoir operation rules? Actually, the authors have conducted an elaborate sensitivity test on the parameters of operation. Some of the combination might have worked as "adaptation" to

[Figure]

climate change.

Figure 1: The figure doesn't have legend. It should be displayed what the height of bars quantitatively indicates.

Figure 5: Specify the base period of these two figures. I'm a bit curious why the plots start form -4% at 2010 (largest change) and gradually "recover" toward 2100 (smallest change) for RCP8.5.

---

## Author Comment (AC1) · 3 May 2016

We are grateful for the thoughtful comments from the reviewer. We write responses to all comments point-by-point as provided below.

Review summary:
This manuscript uses multiple global hydrological models driven by multiple climate model data for two representative concentration pathways (RCPs) to estimate China's hydropower generation potential and the projected future changes based on the river flow estimated by these hydrological models. The study finds that the estimated present-day gross hydropower potential of China is comparable to previous estimates, and suggests that the hydropower potential will decrease in the short-term but will increase by the late 21st century. The study also suggests that these changes vary

significantly across different regions. The results presented are of high interest to the scientific community and beyond as the global society today is increasingly concerned about the use of carbon-intensive energy sources to meet the rising energy demands and hydropower could potentially play an important role in future energy mix toward reducing emissions and mitigating climate change, particularly in the rising economies such as China. Therefore, there is no doubt that the paper addresses an important topic but I feel that the study could be driven more by a central scientific finding with important socio-economic implications, rather than just presenting the changes in hydropower potential across different regions.

**Response**: Thanks for the comments. We have extended the Discussion section with more socioeconomic implications of the hydropower potential changes. The projected hydropower potential changes could be a reference for future hydropower development in China, e.g. to consider the climate change in the estimation of installed hydropower capacity. The increase of GHP in Southwest China may prompt the hydropower development in this region. The decrease of DHP in the hotspot regions implies possible lower power generation from current hydropower facilities. Some technologies, e.g. pumped-storage plants and joint reservoir regulations, may be options to mitigate increased seasonality in streamflow. It highlights the potential need to adapt the reservoir regulations to deal with the likely increasing competitive water uses in future. It should be noted that this study mainly focuses on hydropower potential rather than actual hydropower generation; future hydropower generation is also affected by the energy demand, electricity market, policies, economic conditions, technology development, etc., which are not addressed in the present study. Beyond mitigation and adaptation in river operations, a lower DHP would most likely need to be balanced by energy production from other sources, likely from costlier technologies, implying regional economic impact. Nevertheless, the assessment of hydropower potential in this study can be a fundament for the further investigation of socioeconomic interests of hydropower variation caused by climate change. As a matter of fact, the regional assessment performed in this study provides the necessary information for integration into regional
version of integrated assessment models (IAM) in order to improve their water-energy management and expansion representation while taking into account socio-economic factors for national sustainable mitigation and adaptation policy making.

Specific comments: (1) I suggest the authors to revise the introduction. The first paragraph doesn't read very well. Also, it is important to highlight the objectives of the study and the key questions addressed at the end of introduction.
**Response**: Thanks for the comments. We have revised the introduction and extended the last two paragraphs to further clarify the key questions and objectives in the revised manuscript.

(2) While the gross generation potential provides useful information on the potential future changes, it is not an indicator of actual power generation potential. So, it will be important to consider whether the available flows can be utilized to the fullest as well as various locational and technological constraints. The study doesn't provide any information on this aspect.
**Response**: We agree that the gross hydropower potential (GHP) is far from the actual power generation potential, and have further clarified this in the revised manuscript. Different hydropower potentials exist, namely gross hydropower potential, technical potential, economic potential and exploitable potential (Zhou et al., 2015), which address the multiple constraints of water resources, hydropower technology, economy and environmental protection. However, it is difficult to project the changes of all these potentials (except for GHP, which is constrained only by discharge change) in the future without the use of an integrated assessment model to predict the future economy and technology developments. We focus on the impact of future climate change on the hydropower potential in the present study, and expect to provide a primary reference for the assessment of the impacts of climate change on actual hydropower generation.

Zhou, Y., Hejazi, M., Smith, S., Edmonds, J., Li, H., Clarke, L., Calvin, K. and Thomson, A. (2015) A comprehensive view of global potential for hydro-generated electricity. Energy Environ. Sci., 8(9): 2622-2633.

(3) Moreover, the analysis low flows would provide further insights on how the run-off-the-river hydropower generation capacity would be affected in the future. The annual mean and seasonal changes do not necessarily reflect such effects unless all runoff will be captured in reservoirs.

**Response**: Thanks for the suggestion. We agree with the reviewer that regional results may be affected by the non-representation of the run-of-the-river plants. We did not consider the run-of-the-river stations for lack of hydropower station types in the current database. According to https://en.wikipedia.org/wiki/List_of_run-of-the-river_hydroelectric_power_stations, there are 10 run-of-the-river power plants over 100MW in China presently. The large projects represent a total maximum generating capacity of 4,884 MW, i.e. 2% of the installed 220GW capacity. Many projects under 10MW power plants are not reported within an updated, consistent and exhaustive database across regions. We clarify in the manuscript that we do not take into consideration those run-of-the-river plants. We have added low flow changes in the DHP analysis in the revised manuscript.

(4) In page 4, line 2 it is noted the reservoir module is similar to the one in van Vliet et al. (2016). What are the differences in the findings? It may be worthwhile highlighting the differences.

**Response**: Thanks for the suggestion. We definitely build out of van Vliet et al. (2016) analysis. We specify in the literature review section the scientific gaps between van Vliet et al. and other papers, with respect to the overall objective of the paper, which is to support the sustainable regional development of hydropower in China. In brief, changes of hydropower potentials of reservoirs in China projected by this study and van Vliet et al. (2016) show similar conclusions. According to the objectives of the paper, we complement other's paper findings with the following: First, we used multi-model simulations but van Vliet et al. (2016) used only one global hydrological model which allows us to provide a more exhaustive uncertainty quantification. Secondly, the potential hydropower generation is assessed only over the existing plants in van Vliet et al. (2016), while in this study we assess the hydropower potential generation over different potential development scenarios (installed, gross). Finally, our analysis focuses on regional variability, which is important for development consideration.

(5) Page 5, Line 25: Do all models use the same reservoir operation module?
**Response**: Yes. We specify that the reservoir operations are tuned for each individual reservoir characteristics, i.e. reservoir capacity, and mean annual inflow in particular.

(6) Page 6, Line 24: Change "great" to "high".
**Response**: Changed.

(7) Page 6, Line 12: Why and how were these 447 reservoirs selected?
**Response**: We used as many reservoirs as possible in the study. Those chosen are mostly large reservoirs/dams with key information (i.e. location, storage capacity, dam height) were selected from the GRanD database. This is consistent with other large managed hydrology studies (Hanasaki et al 2008, Döll et al, 2009; van Vliet et al. 2016)

Hanasaki, N., Kanae, S., Oki, T., Masuda, K., Motoya, K., Shirakawa, N., Shen, Y., and Tanaka, K.: An integrated model for the assessment of global water resources – Part 2: Applications and assessments, Hydrol. Earth Syst. Sci., 12, 1027-1037, 10.5194/hess-12-1027-2008, 2008.
Döll, P., Fiedler, K., and Zhang, J.: Global-scale analysis of river flow alterations due to water withdrawals and reservoirs, Hydrol. Earth Syst. Sci., 13, 2413-2432, 10.5194/hess-13-2413-2009, 2009.
van Vliet, M. T. H., Wiberg, D., Leduc, S., and Riahi, K.: Power-generation system vulnerability and adaptation to changes in climate and water resources, Nature Clim. Change, 6, 375-380, 10.1038/nclimate2903, 2016.

(8) Page 7, Line 19: Expand this section or delete this line.
**Response**: Removed.

(9) Section 3: I see that a lot of information is provided as supplementary material. For completeness, I suggest the authors to bring some of these tables to the manuscript

itself.

**Response**: Thanks for the suggestion. We have presented the Table S4, S7 and S10, which show the ensemble means of multimodel, in the revised manuscript.

(10) Page 7, Line 27: change "is" to "are"
**Response**: Corrected.

(11) Page 12, Line 11: Change "great" to "large"
**Response**: Changed.

(12) What is the rationale behind the use of different alpha, beta, and K values? This needs to be discussed in relation to the implications on results.
**Response**: We have clarified the rationale of the sensitivity tests in the revised manuscript. The different values of the parameters in Eq. (1) represent different regulation efficiencies of reservoirs. We performed experiments with different parameter values to show the sensitivity of the results to the regulation coefficients.

(13) Evaporation from water retention behind large dams could increase largely under warmer future climate which can reduce runoff. Is this considered in the present study?
**Response**: Thanks for the comment. We agree that evaporation from reservoir water surface is not negligible and it will be considered in the future work. We did not consider it yet in the present study as the annual evaporation from reservoir surface usually accounts for a relatively small portion of the annual release of large reservoirs (Fekete et al. 2010). According to Liu et al. (2015), evaporation amount from reservoirs is $2.8 \times 10^{10} m^3$ in total, only 0.62% of the total runoff in China and is much smaller than the uncertainty range estimated from the different hydrology models in this study.

Fekete, B. M., Wisser, D., Kroeze, C., Mayorga, E., Bouwman, L., Wollheim, W. M., and Vörösmarty, C.: Millennium Ecosystem Assessment scenario drivers (1970–2050): Climate and hydrological alterations, Global Biogeochem. Cycles, 24, GB0A12, 10.1029/2009GB003593, 2010.

Liu, J., Zhao, D., Gerbens-Leenes, P. W., and Guan, D.: China's rising hydropower demand challenges water sector, Scientific Reports, 5, 11446, 10.1038/srep11446, 2015.

---

## Author Comment (AC2) · 3 May 2016

We are grateful for the thoughtful comments from the reviewer. We write responses to all comments point-by-point as provided below.

General comments
The authors projected Gross Hydropower Potential (GHP) and Developed Hydropower Potential (DHP) of China using the global runoff dataset developed by the ISI-MIP project. The dataset includes global gridded runoff field simulated by 8 global hydrological models for 5 climate models and 2 emission scenarios. They analyzed the spatial and temporal distribution of changes in GHP and DHP in China. Although hydropower is a fundamental source of energy, analyses utilizing macro scale hydrological model have been seldom reported. This report has potential to advance this research field.

[Figure]

As is commonly seen in macro scale hydrological simulations, this study is based on several strong assumptions. I have not been fully convinced by the validity of some of these assumptions. This is partly due to the assumptions themselves, but largely due to lack of discussion. Details are noted below.

First, overall discussion on the background mechanism for the results is lacking. The results the authors obtained are well presented, but why and how they were obtained is little described. The Discussion Section should be largely expanded to include the mechanisms. Second, the term (and the model) of DHP should be revisited. What does "developed potential" mean? Which is more close to hydropower generation or technical hydropower potential? If DHP is different from any important indicators in the real world, how should we interpret the results? Without clarification of DHP, it is not clear what was calculated and what for. Third, the quality in runoff field of ISI-MIP should be well discussed. Since the global hydrological models participated in ISI-MIP have not been calibrated except the WaterGAP model, it must be carefully discussed that how the biases in runoff propagate to the results. Fourth, as far as I understood, the authors assigned the national total Installed Hydropower Capacity (IHC) into 447 major reservoirs. Since this might significantly overestimate IHC at individual reservoirs, the validity of this treatment should be validated and discussed. It might be a good idea to start with comparing reported installed hydropower capacity at individual reservoirs with the authors' estimation.

**Response**: Thanks for the suggestions and comments.

(1) We have extended the description of methods in the sections 2.2 and 2.3 and the Discussion on the interpretation of the results in the revised manuscript (please also refer to the question on Page 11 Line 22).

(2) We have further clarified the term of DHP in the revised manuscript. DHP in this study refers to hydropower potential at the developed plants. The changes of potential of hydropower generation are usually determined by streamflow and hydropower capacity (Lehner et al., 2005). Since we could not predict the development of hydropower technology and capacity, we only present the changes in DHP resulted from

the streamflow variation. We assessed *hydropower potential* in this study generally to highlight the necessity of considering the impact of climate change in hydropower development and planning in China.

(3) We agree that the non-calibrated model data may result in considerable biases in the GHP/DHP estimates. In the revised manuscript, we have extended discussion on the uncertainty of GCM and GHM models to remind the readers of the possible biases and the importance of calibration to the models. Note that we did not use WaterGAP model in this study because the WaterGAP model did not provide daily runoff, which was used for GHP estimation. We have added a brief discussion on the current state of the global hydrological models and the calibrated model such as WaterGAP may show better agreement in the historical period.

(4) We agree with the reviewer that it could be a source of uncertainty and we actually address it. We obtained IHC data at provincial level in China and assigned the IHC values to individual reservoir at each province. We have compared the adjusted IHC with the reported values at some reservoirs and briefly discussed the potential errors in the assignment of IHC values in the manuscript in section 2.3. The adjusted IHCs correspond well to the reported values for the reservoirs that storage capacity is highly related to hydropower capacity; e.g., the relative error is less than 1% for the adjusted IHC of the Three Gorge Reservoir, but is more than 50% for the Gezhouba hydropower station. In the supplemental material, two experiments with different IHC values were performed to show the sensitivity of DHP estimates to the deviation of IHC (Figure S14). Collection and validation of the IHC of individual reservoirs should be important to reduce uncertainty in the DHP estimates in the future work. We now highlight early, in the description of the setup, how this source of uncertainty is addressed later in the discussion and the supplemental material. The discussion already refers to this supplemental material.

Lehner, B., Czisch, G., and Vassolo, S.: The impact of global change on the hydropower potential of Europe: a model-based analysis, Energy Policy, 33, 839-855, 10.1016/j.enpol.2003.10.018, 2005.

Specific comments

Page 6 Line 7," DHP = min (Rm x h x g, IHC)": I found that this equation primarily expresses hydropower generation. Why was this termed "Developed Hydropower Potential", not hydropower generation? If DHP is not hydropower generation, then what is this correspond to in the reality?

**Response**: We emphasized that DHP is a potential because actual hydropower generation is affected by more than discharge and IHC, i.e. energy demand, electricity price, environmental discharge not going through the turbines, etc. It therefore could not correspond to a hydropower production in the actual operations.

Page 6 Line 8 "h=S/A": Fekete et al. (2010) expressed reservoirs as tetrahedrons in their model. What are the advantage and disadvantage of the authors' expression (cylinder)?

**Response**: The cylinder is a simple assumption. It means that in our analysis, we have a linear decrease in head as the reservoir volume decreases. In Fekete et al. (2010), the change in head is slower at first when volume decreases. Based on those simple assumptions, it means that for small to medium changes in inflow, our modeling framework will detect larger changes in DHP than Fekete et al. Beyond an unspecified threshold in decrease in inflow, which will vary for each reservoir, Fekete et al. assumption will be more aggressive and non-linear on the estimate of changes in DHP. The tetrahedrons may be a better approximation for the reservoirs located at the rivers with high stream gradients. It is beyond the scope of this analysis to quantify this uncertainty but we added this discussion in the Discussion section and highlight the differences in DHP assessed in this analysis and in other papers (e.g. van Vliet et al. 2016).

van Vliet, M. T. H., Wiberg, D., Leduc, S., and Riahi, K.: Power-generation system vulnerability and adaptation to changes in climate and water resources, Nature Clim. Change, 6, 375-380, 10.1038/nclimate2903, 2016.

Page 6 line 15 "no IHC data associated with the GRanD reservoirs" World Register of Dam by International Commission of Large Dams (http://www.icold-cigb.org/) includes

Electric Capacity of individual dams.

**Response**: Thank you for the information. IHC is missing in many reservoir entries in the ICOLD database. We added the following statement. "Despite World Register of Dam by International Commission of Large Dams (http://www.icold-cigb.org/) includes Electric Capacity of individual dams, many reservoir entries are missing. Therefore, we used the following approach to represent the IHC at our aggregated reservoirs.

Page 6 line 18 "Then the adjusted provincial IHC..." As far as I understand, this study deals with storage and discharge for 447 reservoirs in China, while IHC for all the nation. This discrepancy could make Rm x h x g substantially smaller than IHC, hence it may have influenced the results. This point should be clarified here.

**Response**: We assigned IHC of provinces (not the nation) to each reservoir according to the storage. The IHC data was collected before 2004, which is close to the GranD database. We have checked that many large reservoirs built in 21st century were not included in the GRanD database. The assignment definitely may bring biases to the DHP estimation (not necessarily smaller than IHC). The experiments with 0.9*IHC and 1.1*IHC should be helpful for addressing the uncertainty resulted from the IHC assignment, and we have further clarified it in the revised manuscript.

Page 11 Line 22. I got a general impression that the Discussion Section is superficial. Since the Results Section only introduces the numbers that authors obtained, actually I expected detailed discussion on the background mechanisms of model behaviors and interpretation of the results, but these are seldom provided in the current form of the manuscript. The contents of this section should be substantially added.

**Response**: Thanks for the suggestion. We have extended the Discussion section in the revised manuscript. The hydropower potential in this study was assessed based on multimodel simulations of runoff and discharge under different climate change scenarios. The assessment of hydropower potential changes is based on the linkages of climate, streamflow and hydropower. Therefore, the projection of streamflow by the GCM-GHM combinations will directly affect the estimation of hydropower potential.

Though the ensemble mean of projected GHP of China for the historical period is relatively close to the reported data, there is large discrepancy among GHMs. During the historical period, discrepancy in hydropower potential is much smaller among GCMs because the GCM climate data is bias-corrected to a historical reference. It implies that validation or bias-correction may be helpful to reduce the uncertainty in the projections of GHMs. However, the GCM uncertainty predominates future GHP changes at most areas in China.

The uncertainty in the streamflow projections also propagate to the estimation of DHP. Though a universal reservoir regulation is applied to all modeled discharge, there is still a large spread across GCM-GHM combinations. The large uncertainty in DHP should be mainly due to the large discrepancy of GCM climate data since the reservoirs used in this study are mostly located in areas with low model agreements in future discharge projections (see Figure 1 in Schewe et al., 2014). This also partly explains why the total DHP (Figure 5) shows larger spread than total GHP (Figure 3) of China.

Schewe, J. et al., 2014. Multimodel assessment of water scarcity under climate change. Proc. Nat. Acad. Sci. U.S.A., 111(9): 3245-3250. DOI:10.1073/pnas.1222460110

Page 12 Line 10 "most regions show poor agreement between models": In terms of what? Magnitude or signs? What are the results of the WaterGAP model or the only model with calibration?

**Response**: The agreement here means signs of the GHP changes. We specified it in the statement. We did not use the WaterGAP model in this study because the WaterGAP model did not provide daily runoff, which was used to estimate the GHP in a routing model at daily step.

Page 13 Line 19 "Thus, reservoir regulation could be changed in the future to adapt to climate change": Too superficial and abstract. How should it be changed based on the findings of this study?

**Response**: We agree with the reviewer. Reservoir regulation rules are related to reservoir functions. In this study, we treated all dams as hydropower stations rather than multi-objective reservoirs. Increase of reservoir release or retain a high water level may produce more DHP. Therefore, DHP could be maximized by adjusting the monthly release, e.g. retaining a high water level seems to be easier to obtain high DHP in the dry season (see Figure S12, where $\beta$ can adjust the proportions of monthly and annual inflow for monthly release). However, considering various competitive water uses, reservoir regulation is optimized for multiple objectives rather than for DHP only. Adaptation of operational reservoir operations to climate change is more complex. We have rewritten this sentence carefully to clarify the findings of this study in the revised manuscript.

Page 15 Line 5 "Relatively small changes also will occur in late spring and early summer, while large decreases will occur in other months". Why did these happen in your simulations? Basic mechanisms should be mentioned here. For instance, DHP is a function of monthly discharge (Rm) and water level (h). Which is dominant factor to produce the seasonal variation?

**Response**: Thanks for the suggestion. We agree with the reviewer that it would be interesting to isolate the drivers of change in DHP. Voisin et al. (2013) describes how generic operating rules affect the reservoir storage, which highlights how monthly release and water level are linked. For the specific release used in this paper (mean annual flow), a large storage capacity reservoir will react to changes in annual mean flow by decreasing its ability to fill, the head will decrease and DHP will decrease. Conversely an increase in flow will top the reservoir during certain years, increase the DHP until reaching a plateau due to the reservoir maximum capacity and induced spilling. Change in the seasonality of the flow will affect the speed at which the reservoir can fill in the Spring, therefore affecting the head. DHP production in Summer are also affected by the level of the reservoir storage on the month when the natural monthly flow is smaller than the mean annual flow (start of the operation season, see Haddeland et al. 2006 and Hanasaki et al. 2006) (see Equation 1). This additional component, which mimics the inter-annual variability in release and operations, will be impacted

by a change in inflow seasonality, possibly affecting the DHP at end of the summer. Drivers in seasonal changes in DHP vary by reservoirs and will overall depends heavily on the simplified representation of reservoir operations. The current assessment also assumes no change in reservoir operations (no adaptation) which affects the seasonal change in DHP. We have added some explanation for the result in the revised manuscript accordingly.

Voisin, N., Li, H., Ward, D., Huang, M., Wigmosta, M. and Leung, L. R. (2013) On an improved sub-regional water resources management representation for integration into earth system models. Hydrol. Earth Syst. Sci., 17(9): 3605-3622.

Page 15 Line 10 "DHPs given the current infrastructure will not be able to mitigate the hydrological changes and thus will decrease": Why and how did the authors conclude this? Would this conclusion be different if the authors modified the reservoir operation rules? Actually, the authors have conducted an elaborate sensitivity test on the parameters of operation. Some of the combination might have worked as "adaptation" to climate change.
**Response**: Thanks for the suggestion. We rewrote this sentence carefully in the revised manuscript. The sensitivity tests to some degree can be regarded as "adaptations" to climate change by modifying regulation coefficients, and this may alter the changes of the hydropower potential of current reservoirs. It should be noted that we do not consider other reservoir purposes in the present study (regulation for irrigation, domestic or other sectorial supply), which may increase competitive water use and then further reduce hydropower generation.

Figure 1: The figure doesn't have legend. It should be displayed what the height of bars quantitatively indicates.
**Response**: Thanks for the suggestion. The bars and texts at the lower left corner of the figure are the legend. We redrew the figure to make it more readable in the revised manuscript.

Figure 5: Specify the base period of these two figures. I'm a bit curious why the plots start form -4% at 2010 (largest change) and gradually "recover" toward 2100 (smallest change) for RCP8.5.

**Response**: The base period of these two figures is 1971-2000. The plots show the temporal changes from 2010-2084, which are 31-year moving averages of the original time serials, e.g. the data of 2010 is the mean value of 1995-2025. For a clear view, we did not show the moving averages of the 1985-2009 period. The discharge is projected to decrease in most areas China during the 2020-2050 period, and significantly recover in some areas during the 2070-2099 period (see Figure S2). The variation of discharge largely affects the DHP variability, however, there are few reservoirs located in the areas with large increase of discharge in the 2070-2099 period and the reservoir regulation may offset the effects of discharge variation to some degree. Therefore, the projected changes of DHP of China would not always coincide with those of discharge. It is not very exact that DHP change recovers toward 2100, but the annual DHP change definitely shows less decrease after 2040 and reach about -1% at late 21st century under RCP2.6.

---

## Author Response (AR1)

**Response to reviewers**

We are grateful for thoughtful comments of the reviewers. We replied to all comments point-by-point as provided below and have revised the manuscript accordingly.

**Anonymous Referee #1**

Review summary:

This manuscript uses multiple global hydrological models driven by multiple climate model data for two representative concentration pathways (RCPs) to estimate China's hydropower generation potential and the projected future changes based on the river flow estimated by these hydrological models. The study finds that the estimated present-day gross hydropower potential of China is comparable to previous estimates, and suggests that the hydropower potential will decrease in the short-term but will increase by the late 21st century. The study also suggests that these changes vary significantly across different regions. The results presented are of high interest to the scientific community and beyond as the global society today is increasingly concerned about the use of carbon-intensive energy sources to meet the rising energy demands and hydropower could potentially play an important role in future energy mix toward reducing emissions and mitigating climate change, particularly in the rising economies such as China. Therefore, there is no doubt that the paper addresses an important topic but I feel that the study could be driven more by a central scientific finding with important socio-economic implications, rather than just presenting the changes in hydropower potential across different regions.

**Response**: Thanks for the review. We address the comment by clarifying the scientific contribution of the analysis (DHP, GHP and multi model analysis) and by adding a discussion section on the socio-economic implications. The projected hydropower potential changes could be a reference for future hydropower development in China, e.g. to consider climate change for expansion of hydropower capacity assets. The increase of GHP in Southwest China may prompt the hydropower development in this region. The decrease of DHP in the hotspot regions implies possible lower power generation from current hydropower facilities. Some technologies, e.g. pumped-storage plants and joint reservoir regulations, may be options to mitigate increased seasonality in streamflow. It highlights the potential need to adapt the reservoir regulations to deal with the likely increasing competitive water uses in future. It should be noted that this study mainly focuses on hydropower potential rather than actual hydropower generation; future hydropower generation is also affected by the energy demand, electricity market, policies, economic conditions, technology development, etc., which are not addressed in the present study. We have extended the Discussion with more socioeconomic implications of the hydropower potential changes (section 4.4, P16L1-14).

Specific comments:

(1) I suggest the authors to revise the introduction. The first paragraph doesn't read very well. Also, it is important to highlight the objectives of the study and the key questions addressed at the end of introduction.

**Response**: Thanks for the comments. We have revised the first paragraph and the last two paragraphs (P3L14–P4L9) in the Introduction to further clarify the key questions and objectives of this study.

(2) While the gross generation potential provides useful information on the potential future changes, it is not an indicator of actual power generation potential. So, it will be important to consider whether the available flows can be utilized to the fullest as well as various locational and technological constraints. The study doesn't provide any information on this aspect.

**Response**: We agree that the gross hydropower potential (GHP) is far from the actual power generation potential, and have further clarified this in the revised manuscript. Different hydropower potentials exist, namely gross hydropower potential, technical potential, economic potential and exploitable potential (Zhou et al., 2015), which address the multiple constraints of water resources, hydropower technology, economy and environmental protection. However, it is difficult to project the changes of all these potentials (except for GHP, which is constrained only by discharge change) in the future without the use of an integrated assessment model to predict the future economy and technology developments. We focus on the impact of future climate change on the hydropower potential in the present study, and expect to provide a primary reference for the assessment of the impacts of climate change on actual hydropower generation.

Zhou, Y., Hejazi, M., Smith, S., Edmonds, J., Li, H., Clarke, L., Calvin, K. and Thomson, A. (2015) A comprehensive view of global potential for hydro-generated electricity. Energy Environ. Sci., 8(9): 2622-2633.

(3) Moreover, the analysis low flows would provide further insights on how the run-off-the-river hydropower generation capacity would be affected in the future. The annual mean and seasonal changes do not necessarily reflect such effects unless all runoff will be captured in reservoirs.

**Response**: Thanks for the suggestion. We agree with the reviewer that regional results may be affected by the non-representation of the run-of-the-river plants. We did not consider the run-of-the-river stations for lack of hydropower station types in the current database. According to https://en.wikipedia.org/wiki/List_of_run-of-the-river_hydroelectric_power_stations, there are 10 run-of-the-river power plants over 100MW in China presently. The large projects represent a total maximum generating capacity of 4,884 MW, i.e. 2% of the installed 220GW capacity. Many projects under 10MW power plants are not reported within an updated, consistent and exhaustive database across regions. We clarified and justified in the manuscript that we did not take into consideration those run-of-the-river plants (P6L21–22).

(4) In page 4, line 2 it is noted the reservoir module is similar to the one in van Vliet et al. (2016). What are the differences in the findings? It may be worthwhile highlighting the differences.

**Response**: Thanks for the suggestion. We leverage van Vliet et al. (2016) analysis. Our objective is to support the sustainable regional development of hydropower in China, which implies a strong regional component to the analysis as well as extensive uncertainty quantification and an additional set of metrics (DHP). We specified the scientific gaps between van Vliet et al. (2016) and other papers in the introduction section (P3L16–20, P4L1–5). Though changes of hydropower potentials of reservoirs in China projected by this study showed

similar result with van Vliet et al. (2016) (P16L20–21), the research we conducted is different with theirs in following aspects. First, we used multimodel simulations but van Vliet et al. (2016) used only one global hydrological model which allows us to provide a more exhaustive uncertainty quantification. Secondly, the potential hydropower generation is assessed only over the existing plants in van Vliet et al. (2016), while in this study we assess the hydropower potential generation over different potential development scenarios (installed, gross). Finally, our analysis focuses on regional variability, which is important for development consideration.

(5) Page 5, Line 25: Do all models use the same reservoir operation module?
**Response**: Yes. We specify that the reservoir operations are tuned for each individual reservoir characteristics, i.e. reservoir capacity, and mean annual inflow associated with the particular GCM used (P6L20–21).

(6) Page 6, Line 24: Change "great" to "high".
**Response**: Changed.

(7) Page 6, Line 12: Why and how were these 447 reservoirs selected?
**Response**: We used as many reservoirs as possible in the study. Those chosen are mostly large reservoirs/dams with key information (i.e. location, storage capacity, dam height) from the GRanD database (P6L19-20). This is consistent with other large scale managed hydrology studies (Hanasaki et al 2008, Döll et al, 2009; van Vliet et al. 2016)

Hanasaki, N., Kanae, S., Oki, T., Masuda, K., Motoya, K., Shirakawa, N., Shen, Y., and Tanaka, K.: An integrated model for the assessment of global water resources – Part 2: Applications and assessments, Hydrol. Earth Syst. Sci., 12, 1027-1037, 10.5194/hess-12-1027-2008, 2008.
Döll, P., Fiedler, K., and Zhang, J.: Global-scale analysis of river flow alterations due to water withdrawals and reservoirs, Hydrol. Earth Syst. Sci., 13, 2413-2432, 10.5194/hess-13-2413-2009, 2009.
van Vliet, M. T. H., Wiberg, D., Leduc, S., and Riahi, K.: Power-generation system vulnerability and adaptation to changes in climate and water resources, Nature Clim. Change, 6, 375-380, 10.1038/nclimate2903, 2016.

(8) Page 7, Line 19: Expand this section or delete this line.
**Response**: Removed.

(9) Section 3: I see that a lot of information is provided as supplementary material. For completeness, I suggest the authors to bring some of these tables to the manuscript itself.
**Response**: Thanks for the suggestion. We have presented the Table S4 and S10, which show the ensemble means of multimodel, in the revised manuscript.

(10) Page 7, Line 27: change "is" to "are"
**Response**: Corrected.

(11) Page 12, Line 11: Change "great" to "large"
**Response**: Changed.

(12) What is the rationale behind the use of different alpha, beta, and K values? This needs to be discussed in relation to the implications on results.
**Response:** We have clarified the rationale of the sensitivity tests in the revised manuscript (P14L5-7). The different values of the parameters in Eq. (1) represent different regulation efficiencies of reservoirs. We performed experiments with different parameter values to show the sensitivity of the results to the regulation coefficients.

(13) Evaporation from water retention behind large dams could increase largely under warmer future climate which can reduce runoff. Is this considered in the present study?
**Response**: Thanks for the comment. We agree that evaporation from reservoir water surface is not negligible and it will be considered in the future work. We did not consider it yet in the present study as the annual evaporation from reservoir surface usually accounts for a relatively small portion of the annual release of large reservoirs (Fekete et al. 2010). According to Liu et al. (2015), evaporation amount from reservoirs is $2.8\times10^{10}m^3$ in total, only 0.62% of the total runoff in China and is much smaller than the uncertainty range estimated from the different hydrology models in this study. (P5L21-22)

Fekete, B. M., Wisser, D., Kroeze, C., Mayorga, E., Bouwman, L., Wollheim, W. M., and Vörösmarty, C.: Millennium Ecosystem Assessment scenario drivers (1970–2050): Climate and hydrological alterations, Global Biogeochem. Cycles, 24, GB0A12, 10.1029/2009GB003593, 2010.
Liu, J., Zhao, D., Gerbens-Leenes, P. W., and Guan, D.: China's rising hydropower demand challenges water sector, Scientific Reports, 5, 11446, 10.1038/srep11446, 2015.

**Anonymous Referee #2**

General comments
The authors projected Gross Hydropower Potential (GHP) and Developed Hydropower Potential (DHP) of China using the global runoff dataset developed by the ISI-MIP project. The dataset includes global gridded runoff field simulated by 8 global hydrological models for 5 climate models and 2 emission scenarios. They analyzed the spatial and temporal distribution of changes in GHP and DHP in China. Although hydropower is a fundamental source of energy, analyses utilizing macro scale hydrological model have been seldom reported. This report has potential to advance this research field.
As is commonly seen in macro scale hydrological simulations, this study is based on several strong assumptions. I have not been fully convinced by the validity of some of these assumptions. This is partly due to the assumptions themselves, but largely due to lack of discussion. Details are noted below.
First, overall discussion on the background mechanism for the results is lacking. The results the authors obtained are well presented, but why and how they were obtained is little described.

The Discussion Section should be largely expanded to include the mechanisms. Second, the term (and the model) of DHP should be revisited. What does "developed potential" mean? Which is more close to hydropower generation or technical hydropower potential? If DHP is different from any important indicators in the real world, how should we interpret the results? Without clarification of DHP, it is not clear what was calculated and what for. Third, the quality in runoff field of ISI-MIP should be well discussed. Since the global hydrological models participated in ISI-MIP have not been calibrated except the WaterGAP model, it must be carefully discussed that how the biases in runoff propagate to the results. Fourth, as far as I understood, the authors assigned the national total Installed Hydropower Capacity (IHC) into 447 major reservoirs. Since this might significantly overestimate IHC at individual reservoirs, the validity of this treatment should be validated and discussed. It might be a good idea to start with comparing reported installed hydropower capacity at individual reservoirs with the authors' estimation.

**Response**: Thanks for the suggestions and comments.

(1) We have revised the Discussion (P13L2-P13L18) to provide more background mechanism on the projected hydropower potential changes (please also refer to the question on Page 11 Line 22).

(2) We have clarified the term of DHP in the revised manuscript (P5L6-8). DHP in this study refers to hydropower potential at the developed plants. The changes of potential of hydropower generation are usually determined by streamflow and hydropower capacity (Lehner et al., 2005). Since we could not predict the development of hydropower technology and capacity, we only present the changes in DHP resulted from the streamflow variation. We assessed *hydropower potential* in this study generally to highlight the necessity of considering the impact of climate change in hydropower development and planning in China.

(3) We agree that the non-calibrated model data may result in considerable biases in the GHP/DHP estimates. In the revised manuscript, we have extended discussion on the uncertainty of GCM and GHM models to remind the readers of the possible biases and the importance of calibration to the models (P12L10-L25, P13L30-32). Note that we did not use WaterGAP model in this study because the WaterGAP model did not provide daily runoff, which was used for GHP estimation. We have highlighted the importance of calibration and validation (as the WaterGAP model did) to the current global hydrological models, which may show better agreement in the multimodel simulations.

(4) We agree with the reviewer that it could be a source of uncertainty and we actually address it. We obtained IHC data at provincial level in China and assigned the IHC values to individual reservoir at each province. We compared the adjusted IHC with the reported values at some reservoirs and briefly discussed the potential errors in the assignment of IHC values in the manuscript in section 2.3 (P6L29-32). The adjusted IHCs correspond well to the reported values for the reservoirs that storage capacity is highly related to hydropower capacity; e.g., the relative error is less than 1% for the adjusted IHC of the Three Gorge Reservoir, but is more than 50% for the Gezhouba hydropower station. In the supplemental material, two experiments with different IHC values were performed to show the sensitivity of DHP estimates to the deviation of IHC (P14L13-17, Figure S14). Collection and validation of the IHC of individual reservoirs should be important to reduce uncertainty in the DHP estimates in the future work. We now highlight early, in the description of the setup, how this source of uncertainty is addressed later

in the discussion and the supplemental material. The discussion already refers to this supplemental material.

(5) In the Discussion section (4.4, P16L1-14), we also added how this assessment is to be used, i.e. for implementation with large scale integrated assessment model in order to evaluate the socio-economic development of China, or as first assessment for developers to spot potential sites for additional analysis.

Lehner, B., Czisch, G., and Vassolo, S.: The impact of global change on the hydropower potential of Europe: a model-based analysis, Energy Policy, 33, 839-855, 10.1016/j.enpol.2003.10.018, 2005.

Specific comments

Page 6 Line 7," DHP = min (Rm x h x g, IHC)": I found that this equation primarily expresses hydropower generation. Why was this termed "Developed Hydropower Potential", not hydropower generation? If DHP is not hydropower generation, then what is this correspond to in the reality?

**Response**: We emphasized that DHP is a potential because actual hydropower generation is affected by more than discharge and IHC, i.e. energy demand, electricity price, environmental discharge not going through the turbines, etc. It therefore could not correspond to a hydropower production in the actual operations.

Page 6 Line 8 "h=S/A": Fekete et al. (2010) expressed reservoirs as tetrahedrons in their model. What are the advantage and disadvantage of the authors' expression (cylinder)?

**Response**: The cylinder is a simple assumption. Based on this assumption, the hydraulic head changes linearly as the reservoir volume changes, while in Fekete et al. (2010) the head show nonlinear changes. We perform simple experiments to show the different changes in hydraulic head with different simplification of the reservoir geometry (Figure 1). For the experiment (a), it shows that for small changes in inflow, both the cylinder and tetrahedron assumptions will detect small changes in head. Beyond an unspecified threshold in decrease (increase) in inflow, which may vary for each reservoir, Fekete et al. assumption will be more (less) aggressive and non-linear on the estimate of changes in DHP. For the experiment (b), the tetrahedron assumption would show less changes in hydraulic head, nearly 0.4 of those for the cylinder assumption, because the surface area of the tetrahedron is much larger (about twofold) than the cylinder. The tetrahedrons may be a better approximation for the reservoirs located in rivers with high stream gradients. Therefore, the different assumption of the reservoir geometry would result in different hydropower potential estimates (P6L13-L17). It is beyond the scope of this analysis to quantify this uncertainty but we added brief discussion in the Discussion section (P14L19-24).

[Figure]

Figure 1. The head changes under different assumptions of the reservoir geometry (cylinder and tetrahedron). The experiment is conducted for the case that the cylinder and the tetrahedron have the same volume and (a) the same surface area, and (b) the same water depth at the beginning. The head changes are computed for the inflow changes by -50% to 50%. The expression of Fekete et al. (2010) is adopted to compute the storage change of tetrahedron.

Page 6 line 15 "no IHC data associated with the GRanD reservoirs" World Register of Dam by International Commission of Large Dams (http://www.icold-cigb.org/) includes Electric Capacity of individual dams.

**Response**: Thank you for the information. IHC is missing in many reservoir entries in the ICOLD database. We added the following statement. Despite World Register of Dam by International Commission of Large Dams (http://www.icold-cigb.org/) includes Electric Capacity of individual dams, many reservoir entries are missing. Therefore, we used the following approach to represent the IHC at our aggregated reservoirs.

Page 6 line 18 "Then the adjusted provincial IHC…" As far as I understand, this study deals with storage and discharge for 447 reservoirs in China, while IHC for all the nation. This discrepancy could make Rm x h x g substantially smaller than IHC, hence it may have influenced the results. This point should be clarified here.

**Response**: We assigned IHC of provinces (not the nation) to each reservoir according to the storage. The IHC data was collected before 2004, which is close to the GranD database. We have checked that many large reservoirs built in 21$^{st}$ century were not included in the GRanD database. The assignment definitely may bring biases to the DHP estimation (not necessarily smaller than IHC). The experiments with 0.9*IHC and 1.1*IHC should be helpful for addressing the uncertainty resulted from the IHC assignment, and we have further clarified it in the revised manuscript (P6L29-32).

Page 11 Line 22. I got a general impression that the Discussion Section is superficial. Since the Results Section only introduces the numbers that authors obtained, actually I expected detailed discussion on the background mechanisms of model behaviors and interpretation of the results,

but these are seldom provided in the current form of the manuscript. The contents of this section should be substantially added.

**Response**: Thanks for the suggestion. We have extended the Discussion section in the revised manuscript (P12L10-P13L18). The hydropower potential in this study was assessed based on multimodel simulations of runoff and discharge under different climate change scenarios. The assessment of hydropower potential changes is based on the linkages of climate, streamflow and reservoir storage. Therefore, the projection of streamflow by the GCM-GHM combinations will directly affect the estimation of hydropower potential. Though the ensemble mean of projected GHP of China for the historical period is relatively close to the reported data, there is large discrepancy among GHMs. During the historical period, discrepancy in hydropower potential is much smaller among GCMs because the GCM climate data is bias-corrected to a historical reference. It implies that validation or bias-correction may be helpful to reduce the uncertainty in the projections of GHMs. Currently most GHMs are not calibrated against historical observations and often show a large uncertainty in streamflow projections (Schewe et al., 2014). For annual estimates, it should be more effective and important to enhance the middle- and long-term hydrological prediction in order to fine tune the estimates of DHP and GHP. Therefore, validation and calibration of GHMs with hydrological observations (as the WaterGAP model did) are necessary in future studies which are effective in narrowing the differences among GHMs (Müller Schmied et al., 2014; Döll et al., 2016).

As stated, the uncertainty in the streamflow projections certainly propagates to the estimation of DHP. Though a universal reservoir regulation is applied to all modelled discharge, there is still a large spread across GCM-GHM combinations. The large uncertainty in DHP should be mainly due to the large discrepancy of GCM climate data since the reservoirs used in this study are mostly located in areas with low model agreements in future discharge projections (see Figure 1 in Schewe et al., 2014). This also partly explains why the total DHP (Figure 5) shows somewhat larger spread than the total GHP (Figure 3) of China.

Page 12 Line 10 "most regions show poor agreement between models": In terms of what? Magnitude or signs? What are the results of the WaterGAP model or the only model with calibration?

**Response**: The agreement here means signs of the GHP changes. We specified it in the statement (P13L29-30). We did not use the WaterGAP model in this study because the WaterGAP model did not provide daily runoff, which was used to estimate the GHP in a routing model at daily step.

Page 13 Line 19 "Thus, reservoir regulation could be changed in the future to adapt to climate change": Too superficial and abstract. How should it be changed based on the findings of this study?

**Response**: We agree with the reviewer. Reservoir regulation rules are related to reservoir functions. In this study, we treated all dams as hydropower stations rather than multi-objective reservoirs. Increase of reservoir release or retain a high water level may produce more DHP. Therefore, DHP could be maximized by adjusting the monthly release, e.g. retaining a high water level seems to be easier to obtain high DHP in the dry season (see Figure S12, where $\beta$ can adjust the proportions of monthly and annual inflow for monthly release). However,

considering various competitive water uses, reservoir regulation is optimized for multiple objectives rather than for DHP only. Adaptation of operational reservoir operations to climate change is more complex. We have rewritten this sentence carefully to clarify the findings of this study (P14L14-17).

Page 15 Line 5 "Relatively small changes also will occur in late spring and early summer, while large decreases will occur in other months". Why did these happen in your simulations? Basic mechanisms should be mentioned here. For instance, DHP is a function of monthly discharge (Rm) and water level (h). Which is dominant factor to produce the seasonal variation?

**Response**: Thanks for the suggestion. We agree with the reviewer that it would be interesting to isolate the drivers of change in DHP. Voisin et al. (2013) described how generic operating rules affect the reservoir storage, and highlighted how monthly release and water level are linked. For the specific release used in this paper (mean annual flow), a reservoir with large storage capacity will react to changes in annual mean flow by decreasing its capacity, then the head will decrease and DHP will decrease. Conversely an increase in flow will top the reservoir during certain years, increase the DHP until reaching a plateau due to the reservoir maximum capacity and induced spilling. Change in the seasonality of the flow will affect the filling speed in the spring, therefore affecting its head. DHP production in Summer are also affected by the level of the reservoir storage on the month when the natural monthly flow is smaller than the mean annual flow (start of the operation season, see Haddeland et al. 2006 and Hanasaki et al. 2006) (see Equation 1). This additional component, which mimics the inter-annual variability in release and operations, will be impacted by a change in inflow seasonality, possibly affecting the DHP at end of the summer. Drivers in seasonal changes in DHP vary by reservoirs and will overall depends heavily on the simplified representation of reservoir operations. The current assessment also assumes no change in reservoir operations (no adaptation), which affects the seasonal change in DHP. We have added a brief explanation for the result in the revised manuscript accordingly (P17L11-13).

Voisin, N., Li, H., Ward, D., Huang, M., Wigmosta, M. and Leung, L. R. (2013) On an improved sub-regional water resources management representation for integration into earth system models. Hydrol. Earth Syst. Sci., 17(9): 3605-3622.

Page 15 Line 10 "DHPs given the current infrastructure will not be able to mitigate the hydrological changes and thus will decrease": Why and how did the authors conclude this? Would this conclusion be different if the authors modified the reservoir operation rules? Actually, the authors have conducted an elaborate sensitivity test on the parameters of operation. Some of the combination might have worked as "adaptation" to climate change.

**Response**: Thanks for the suggestion. We rewrote this sentence carefully in the revised manuscript (P17L20-22). The sensitivity tests to some degree can be regarded as "adaptations" to climate change by modifying regulation coefficients, and this may alter the changes of the hydropower potential of current reservoirs. It should be noted that we do not consider other reservoir purposes in the present study (regulation for irrigation, domestic or other sectorial supply), which may increase competitive water use and then further reduce hydropower generation.

Figure 1: The figure doesn't have legend. It should be displayed what the height of bars quantitatively indicates.

**Response**: Thanks for the suggestion. The bars and texts at the lower left corner of the figure are the legend. We redrew the figure to make it more readable in the revised manuscript.

Figure 5: Specify the base period of these two figures. I'm a bit curious why the plots start form -4% at 2010 (largest change) and gradually "recover" toward 2100 (smallest change) for RCP8.5.

**Response**: The base period of these two figures is 1971-2000. The plots show the temporal changes from 2010-2084, which are 31-year moving averages of the original time serials, e.g. the data of 2010 is the mean value of 1995-2025. For a clear view, we did not show the data of the 1985-2009 period. The discharge is projected to decrease in most areas China during the 2020-2050 period, and significantly recover in some areas during the 2070-2099 period (see Figure S2). The variation of discharge largely affects the DHP variability, however, there are few reservoirs located in the areas with large increase of discharge in the 2070-2099 period and the reservoir regulation may offset the effects of discharge variation to some degree. Therefore, the projected changes of DHP of China would not always coincide with those of discharge. It is not very exact that DHP change recovers toward 2100, but the annual DHP change definitely shows less decrease after 2040 and reach about -1% at late 21$^{st}$ century under RCP2.6.

[revised manuscript text omitted]